# Vision impairment and associated daily activity limitation: A systematic review and meta-analysis

Masoud Rahmati[1,2,3]*, Lee Smith[4], Laurent Boyer[1], Guillaume Fond[1], Dong Keon Yon[5,6], Hayeon Lee[5], Tarnjit Sehmbi[7], Mapa Prabhath Piyasena[7], Shahina Pardhan[7,8]*

1 AP-HM, Research Centre on Health Services and Quality of Life, Aix Marseille University, Marseille, France, 2 Department of Physical Education and Sport Sciences, Faculty of Literature and Human Sciences, Lorestan University, Khoramabad, Iran, 3 Department of Physical Education and Sport Sciences, Faculty of Literature and Humanities, Vali-E-Asr University of Rafsanjan, Rafsanjan, Iran, 4 Centre for Health, Performance, and Wellbeing, Anglia Ruskin University, Cambridge, United Kingdom, 5 Center for Digital Health, Medical Science Research Institute, Kyung Hee University Medical Center, Kyung Hee University College of Medicine, Seoul, Republic of Korea, 6 Department of Pediatrics, Kyung Hee University College of Medicine, Seoul, Republic of Korea, 7 Vision and Eye Research Institute, School of Medicine, Anglia Ruskin University, Cambridge, United Kingdom, 8 Centre for Inclusive Community Eye Health, Anglia Ruskin University, Cambridge, United Kingdom

* masoud.rahmati@univ-amu.fr (MR); shahina.pardhan@aru.ac.uk (SP)

**Data Availability Statement:** All relevant data are within the manuscript and its Supporting Information files.

## Abstract

### Background

Vision impairment is a common disability that poses significant challenges to individuals' ability to perform activities essential for independent living, including activities of daily living (ADL) and instrumental activities of daily living (IADL). Despite extensive research, the extent and nature of these associations remain unclear, particularly across varying levels and types of vision impairment.

### Objectives

This meta-analysis aims to estimate associations between vision impairment and difficulties with ADL and IADL.

### Methods

We conducted a systematic review of relevant literature from the inception of the databases to February 2024, using electronic database searches, including PubMed, MEDLINE (Ovid), EMBASE, Cochrane CENTRAL, and CDSR. The articles were screened for title and abstract and then for the full-text reports by two independent reviewers and study quality was appraised. Meta-analyses were performed using random effects models to calculate the pooled effect size, expressed as odds ratio (OR) with corresponding 95% confidence interval (CI) of each outcome.

**Funding:** This study was supported by Vision and Eye Research Institute, School of Medicine, Anglia Ruskin University, Young Street, Cambridge, United Kingdom. The funder had no role in the design and conduct of the study; collection, management, analysis and interpretation of the data; preparation, review or approval of the manuscript; or decision to submit the manuscript for publication.

**Competing interests:** The authors have declared that no competing interests exist.

## Results

Forty-six studies involving 210,960 participants were included. A positive large correlation between vision impairment and difficulties with ADL (Correlation coefficient [*r*] = 0.55, 95% CI 0.37–0.68, *p* = 0.001) and IADL (*r* = 0.60, 95% CI 0.49–0.69, *p* = 0.001) was shown. We also found that vision impairment was associated with difficulties in ADL (OR = 1.77, 95% CI 1.56–2.01, *p* < 0.0001) and IADL (OR = 1.96, 95% CI 1.68–2.30, *p* < 0.0001). Subgroup analysis revealed that moderate to severe impairment resulted in difficulties in ADL (OR = 1.78, 95% CI 1.43–2.21, *p* = 0.02) and IADL (OR = 1.86, 95% CI 1.57–2.20, *p* = 0.0003). Further, there was a significant association between mild to moderate vision impairment and difficulties in IADL (OR = 1.38, 95% CI 1.23–1.55, *p* < 0.0001). Greater impact was observed in individuals with near vision impairment compared to those with distance vision impairment. Near vision impairment was significantly associated with higher difficulties in ADL (OR = 1.77, 95% CI 1.57–2.01, *p* < 0.0001) and IADL (OR = 1.79, 95% CI 1.32–2.42, *p* < 0.0001). In contrast, distance vision impairment showed lower but still significant associations with IADL (OR = 1.19, 95% CI 1.05–1.34, *p* = 0.005) and a nonsignificant association with ADL (OR = 1.12, 95% CI 0.90–1.40, *p* = 0.30). Meta-regression analysis indicated that for every one-year increase in age, ADL performance decreased by an average of 0.0147 units (p < 0.001), while IADL performance declined at a slower rate of 0.0047 units/year (p = 0.031).

## Conclusion

The present systematic review and meta-analysis using several statistical methods indicates that vision impairment including near vision impairment, is associated with difficulties in ADL and IADL. Thus, vision impairment remains an urgent and increasingly important public health priority. These findings highlight the need for targeted measures to raise public health awareness to provide rehabilitation and eye care examination strategies to reduce the risk of developing disabilities in adults and the elderly who have vision impairment.

## 1. Introduction

Vision impairment and blindness are common disabilities affecting more than 338.3 million people worldwide, and that their prevalence increases with advancing age [1, 2]. It has been predicted that the prevalence of vision impairment and blindness will more than double over the next 30 years [1, 2]. Vision impairment is associated with functional disability including activities of daily living, an increased risk of falls, cognitive impairment and dementia, depression, disability, loss of independence, and mortality [1–4]. Near vision impairment or presbyopia is also an important domain in visual disability affecting activities of daily living and there are 1.8 billion people globally with presbyopia [3].

Activities of daily living (ADLs), as an essential component of healthy aging, refer to the fundamental skills necessary for daily self-care. These are further categorized into basic ADL and instrumental ADL (IADL) [5]. ADL encompasses fundamental skills typically needed to manage basic physical needs including feeding, personal hygiene, dressing, ambulating, continence, and toileting. IADL includes more complex activities and organizational skills related to independent living in the community such as housekeeping, managing finances, handling

medications, and meal preparation [5, 6]. The ability to perform ADLs and IADLs without any difficulties is dependent upon cognitive, motor, and perceptual abilities [5] as well as sensory capability. Accordingly, several studies have reported an association between vision impairment and difficulties in ADL and IADL [7–14]. While various individual studies addressed the association between vision impairment and difficulties with ADL and IADL, there is no systematic review and meta-analysis to summarize the pool effects of available evidence. An improved understanding of the association between vision impairment and difficulties in ADL and IADL is needed to inform public policy, public health planning, and allocation of limited healthcare resources. Therefore, we conducted a systematic review and meta-analysis to summarize the current evidence on the association between vision impairment and difficulties with ADL and IADL.

## 2. Methods

The present systematic review and meta-analysis adhered to the methodological guidelines from the Cochrane Handbook for Systematic Reviews and followed the PRISMA (Preferred Reporting Items for Systematic Review and Meta-Analyses) statement 2020 in conducting and reporting the review [15]. This systematic review was pre-registered with the International Prospective Register of Systematic Reviews (PROSPERO; ref. no. CRD42023490518). The PRISMA checklist and Meta-analysis of Observational Studies in Epidemiology (MOOSE) checklist [16] are provided, respectively, in S1 and S2 Checklists.

### 2.1. Search strategy

Two researchers (MR and DKY) electronically searched four databases, including PubMed, MEDLINE (Ovid), EMBASE, Cochrane CENTRAL and CDSR from inception of databases up to February 2024 and disagreements were resolved through discussion with a third reviewer (Sh. P). The search strategy and terms are provided in S1 Table. To find all eligible articles, we searched all reference lists of included studies related to the research question and no language restrictions for studies with English summary were applied in our systematic search.

### 2.2. Eligibility criteria

The present systematic review and meta-analysis adhered to the inclusion criteria according to the PICO criteria [17]. PICO: Participants include people with vision impairment; Outcome includes those studies reporting difficulties in ADL and IADL; Comparison includes people with normal vision; Intervention is not applicable in the present study. We included both prospective and retrospective cohorts, and cross-sectional studies that evaluated the risk of developing disability in ADL and IADL in participants with vision impairment (S2 Table). We excluded studies lacking data to calculate odds ratio or association between vision impairment with disability in ADL or IADL. Studies were excluded if their primary research question was not exploring the association between vision impairment with disability in ADL or IADL. Additionally, studies were excluded if they were narrative literature reviews (although their reference lists were explored for potentially eligible studies; S3 Table).

### 2.3. Data extraction and quality assessment

We extracted data using Covidence systematic review software (version 2, Veritas Health Innovation, Melbourne, VIC, Australia) on a pre-designed spreadsheet, following Cochrane guidelines. The following data were extracted from the eligible studies: author and year, study design, country, age of participants, sample size, the proportion of female participants, ADL

and IADL measurements, vision assessment criteria, vision impairment characteristics, and adjusted variables. The primary outcome was the association between vision impairment with disability in ADL. The secondary outcome was the association between vision impairment with disability in IADL. The quality of included prospective studies were assessed using the Newcastle–Ottawa Scale (NOS) [15, 18, 19]. Data extraction and quality assessment were independently performed by two reviewers (MR and DKY), and disagreements were resolved through discussion with a third reviewer (Sh. P) before conducting meta-analysis.

## 2.4. Statistical analyses

Outcomes were pooled and expressed as odds ratio (OR) with corresponding 95% confidence intervals (CI) based on one-stage approach and the random-effects estimate using the DerSimonian-Laird method [20, 21]. When data extraction for re-estimation of the association between vision impairment and ADL or IADL was not possible, study reported estimates (log- odds ratio) and variances were combined directly using generic inverse variance meta-analysis [20, 22, 23]. To evaluate the potential impact of age on the relationship between vision impairment and ADL or IADL, a random-effects meta-regression analysis was conducted. The dependent variable was the Fisher z-transformed correlation coefficient, and age was used as the moderator (independent variable) in the analysis, employing the restricted maximum likelihood (REML) approach. Meta-analyses of correlations across observational studies were carried out where the relationship between vision impairment with ADL and IADL scores were measured using the same constructs. A Fisher $z$ transformation of the correlation coefficient was carried out, and random-effects meta-analysis of the transformed values was conducted. Pearson $r$ values of 0.1, 0.3, and 0.5 were considered to show small, moderate, and large effects, respectively, and are presented with 95% CIs [24]. MedCalc software version 20.104 (MedCalc software Ltd, Acacialaan 22 8400 Ostend-Belgium) was used to perform meta-analysis of correlational data [25]. The degree of between-study heterogeneity that could not be ascribed to sampling error was explored using Cochran's Q statistics and I-squared ($I^2$; low: 0–40%%, moderate: 30–60%, substantial: 50–90%, and considerable: > 75%) to estimate heterogeneity. Further, the potential for publication bias was assessed using funnel plots with *Egger's* linear regression and *Begg's* rank tests, when the sufficient number of studies (n > 10) was available [21, 26]. Finally, to assess the robustness of summary estimates and to detect if any particular study accounted for a large proportion of heterogeneity, sensitivity analysis was performed by the leave-one-out method [6, 27]. All meta-analyses in the current study were conducted using Review Manager (version 5.4; The Nordic Cochrane Centre, Copenhagen, Denmark), MedCalc software version 20.104 (MedCalc software Ltd, Acacialaan 22 8400 Ostend-Belgium), and Comprehensive Meta-analysis (version 3.3; Biostat Inc., Englewood, NK), a two-sided P value less than 0.05 was considered statistically significant.

## 2.5. Subgroup analysis

We performed four sets of subgroup analyses by 1) different vision impairment assessment (self-reported versus objectively measured), 2) severity of vision impairment ((mild to moderate [visual acuity between 20/200-20/70 in the better seeing eye]) versus moderate to severe (visual acuity between 20/70 to 20/160 and worse than 20/200 in the better seeing eye)), 3) different ADL or IADL assessment (self-reported versus objectively measured by a trained neuropsychologist or a registered nurse), and 4) different vision impairment characteristics (distance vision versus near vision and both distance and near vision).

## 3. Results

### 3.1. Study identification and characteristics

A total of 3304 titles were identified through database searches. 1837 studies remained after removing duplicates. After screening titles and abstracts, 1756 research articles were excluded. Of 81 obtained research articles, another 35 articles were excluded (other outcomes considered (n = 31), case study (n = 2), and reviews (n = 2)). Finally, 46 articles met the eligibility criteria and were included in the meta-analysis (Fig 1). Included studies were published between 1994 to 2022. A total of 210,960 participants were included in this analysis. The age of the participants ranged from 18 to 105 years.

Included studies used the following assessment criteria and charts to detect vision impairment: Self-reported data (N = 17) [7, 8, 12, 14, 28–40], ophthalmologists examination (N = 9) [9, 41–48], Snellen E Chart (N = 3) [10, 49, 50], Bailey–Lovie chart (N = 3) [51–53], Monoyer chart [54], Lighthouse near VA chart (N = 2) [11, 55], Pelli-Robson CS chart (N = 3) [11, 48, 51], Parinaud scale [56], Berkeley glare test [57], Early Treatment Diabetic Retinopathy Study charts (ETDRS) (N = 3) [13, 46, 47], Visual acuity criterion of legal blindness in the

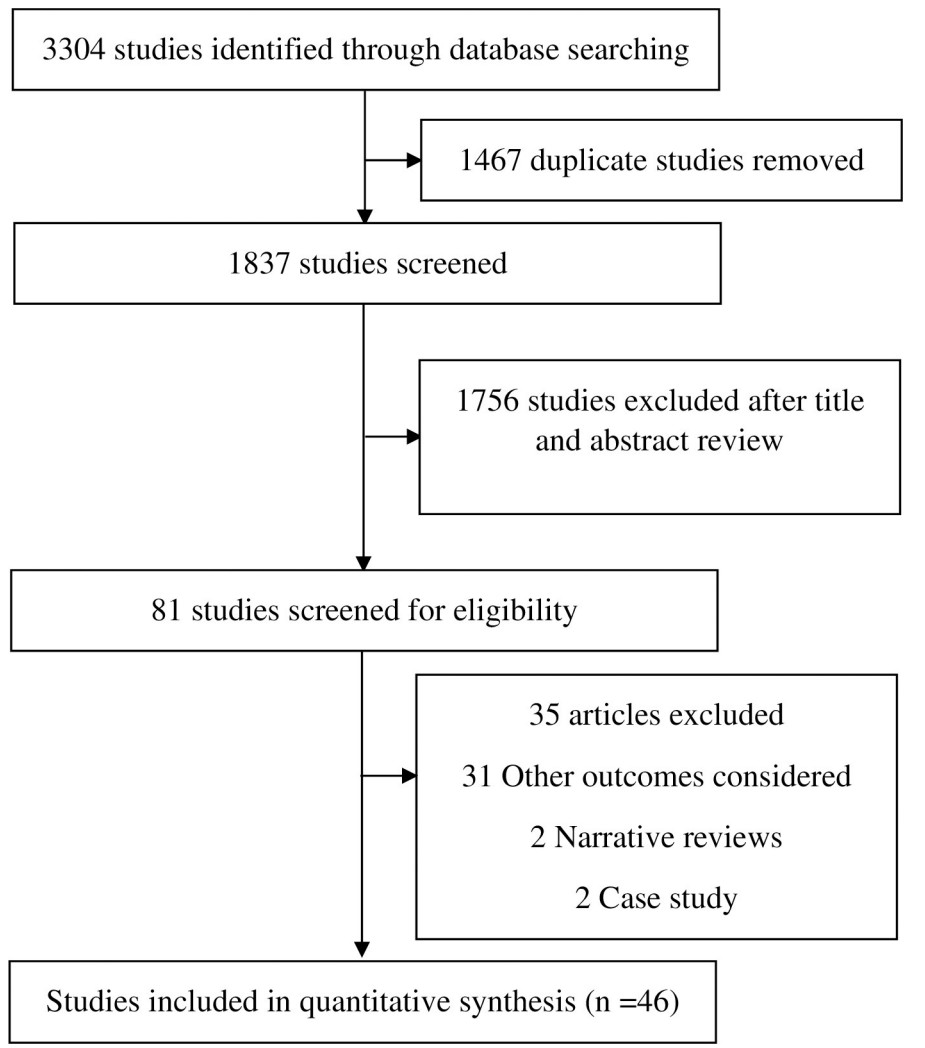

**Fig 1. PRISMA flow diagram of study selection.**

United States [58], Visual Analogue Scale [14], Vistech VCTS 6500 charts [52], Physician diagnosis [59], ability to see a break in a circle on a cardboard sheet 1m away [60], Randot Circles chart [53], International Classification of Functioning, Disability and Health [61], N-30–5 algorithm of FDT perimetry [62], Humphrey 81-point full-field screen [48], and RANDOT circles test [48]. Vision impairment characteristics were not reported in 23 studies [7–9, 12, 14, 28–31, 33–40, 52, 53, 57, 61–63]. Although, visual acuity worse than 0.3 logarithm of the minimum angle of resolution (logMAR) (N = 10) [10, 41, 43–46, 48–50, 56] and worse than 0.5 logMAR (N = 7) [13, 42, 46, 49, 55, 58, 64] in the better-seeing eye were reported in some studies. Additionally, vision impairment characteristics in four studies were reported based on common eye disorders and diseases [47, 51, 54, 59].

Katz ADL was used in 20 studies [10, 31, 32, 34, 35, 37–40, 43, 44, 47, 48, 50, 53–55, 57, 60, 62] and Barthle index was used in three studies [8, 9, 14] respectively to assess difficulties with ADL. Lawton instrumental activities of daily living was used in 25 studies to assess difficulties with IADL [10–12, 31–35, 37–40, 43–45, 47–50, 53–55, 57, 60, 62]. All other studies had developed an adapted questionnaire to assess the motor component of ADL and IADL. Activity limitation in ADL or IADL were actively screened by a trained neuropsychologist or a registered nurse during an at-home visit in six studies [43, 44, 52, 54–56].

All excluded studies are listed in S3 Table. Included studies were of cohort (N = 21) [13, 14, 28–34, 37–40, 42, 44, 48, 49, 56, 58, 60, 61] and cross-sectional design (N = 25) [7–12, 35, 36, 41, 43, 45–47, 50–57, 59, 62–64] and were of medium to high quality, with NOS scores between 5 and 9 (S4 Table). Quantitative analysis of publication bias with Egger's test and Begg's test was non-significant for all analyses (S1 Fig). The general characteristics of included studies are provided in Table 1.

**Table 1. General characteristics of included studies.**

| Study | Design | Country | Age (year) | Participant (Female%) | ADL measurement | IADL measurement | Vision assessment | Vision impairment characteristics | Adjusted variables |
|-------|--------|---------|-----------|----------------------|-----------------|------------------|-------------------|-----------------------------------|--------------------|
| Bekibele et al. 2008 [32] | Cohort | Nigeria | >65 | 2054 (53) | Katz Activities of Daily Living | Nagi Physical Performance Scale and the Health Assessment Questionnair | Self-reported vision impairment | Distance vision Near vision | Socio-demographic Health-related |
| Berger et al. 2008 [7] | Cross-sectional | US | >65 | 9115 (NR) | Difficulty with getting across a room, dressing, bathing, eating, getting out of bed, or using the toilet | Difficulty with preparing a hot meal; shopping for groceries; making phone calls; taking medications; and managing money | Self-reported vision impairment | NR | Socio-demographic Health-related |
| Bouscaren et al. 2019 [33] | Cohort | France | >75 | 4010 (100) | NR | Lawton instrumental activities of daily living | Self-reported vision impairment | NR | Socio-demographic Health-related |
| Brennan et al. 2005 [34] | Cohort | US | >70 | 5151 (NR) | Katz Activities of Daily Living | Lawton instrumental activities of daily living | Self-reported vision impairment | NR | Socio-demographic Health-related |
| Cacciatore et al. 2004 [35] | Cross-sectional | Italy | >65 | 1780 (57) | Katz Activities of Daily Living | Lawton instrumental activities of daily living | Self-reported vision impairment | NR | Socio-demographic Health-related |

(*Continued*)

**Table 1.** (Continued)

| Study | Design | Country | Age (year) | Participant (Female%) | ADL measurement | IADL measurement | Vision assessment | Vision impairment characteristics | Adjusted variables |
|---|---|---|---|---|---|---|---|---|---|
| Cao et al. 2021 [60] | Cohort | China | >65 | 16151 (51) | Katz Activities of Daily Living | Lawton instrumental activities of daily living | Ability to see a break in a circle on a cardboard sheet 1m away | NR | Socio-demographic Health-related |
| Chan et al. 2021 [8] | Cross-sectional | Malaysia | >60 | 3977 (53) | Barthel index | NR | Washington Group Extended Questions Set on Functioning | NR | Socio-demographic Health-related Social-related |
| Cimarolli et al. 2014 [36] | Cross-sectional | US | >95 | 119 (78) | Older Americans Resources and Services Multidimensional Functional Assessment Questionnaire | Older Americans Resources and Services Multidimensional Functional Assessment Questionnaire | Self-reported vision impairment | NR | Socio-demographic Health-related |
| Crews et al. 2004 [28] | Cohort | National Center for Health Statistics | >70 | 6379 (65) | Difficulty with getting across a room, dressing, bathing, eating, getting out of bed, or using the toilet | Difficulty with preparing a hot meal; shopping for groceries; making phone calls; taking medications; and managing money | Self-reported vision impairment | NR | Socio-demographic Health-related |
| Daien et al. 2014 [41] | Cross-sectional | France | >65 | 1887 (55) | NR | Lawton instrumental activities of daily living | Ophthalmologists | Mild: 0.3–0.5 (20/40–20/70) Moderate to severe: worse than 0.5 <20/70 | Socio-demographic Health-related |
| Dargent-Molinaet al. 1996 [49] | Cohort | France | >75 | 1210 (100) | NR | Lawton instrumental activities of daily living | Snellen E Chart | Corrected acuity 5-7/10 3-4/10 ≥ 2/10 | Socio-demographic Health-related |
| Dijkhuizen et al. 2016 [9] | Cross-sectional | Netherland | 19–86 | 240 (37) | Barthel index Comfortable Walking Speed | NR | Ophthalmologists | NR | Socio-demographic Health-related |
| Ensrud et al. 1994 [53] | Cross-sectional | US | >65 | 9704 (100) | Katz Activities of Daily Living | Lawton instrumental activities of daily living | Bailey–Lovie chart Randot Circles chart Vistech Contrast Sensitivity Test System | NR | Socio-demographic Health-related |
| Falahaty et al. 2015 [10] | Cross-sectional | Malaysia | >60 | 150 (54) | Katz Activities of Daily Living | Lawton instrumental activities of daily living | Snellen E Chart | Corrected acuity 6/18-6/36 | Socio-demographic Health-related |
| Grue et al. 2009 [29] | Cohort | Denmark | >75 | 770 (65) | Difficulty with getting across a room, dressing, bathing, eating, getting out of bed, or using the toilet | Difficulty with preparing a hot meal; shopping for groceries; making phone calls; taking medications; and managing money | Unable to read regular print in a newspaper | NR | Socio-demographic Health-related |

*(Continued)*

**Table 1.** (Continued)

| Study | Design | Country | Age (year) | Participant (Female%) | ADL measurement | IADL measurement | Vision assessment | Vision impairment characteristics | Adjusted variables |
|---|---|---|---|---|---|---|---|---|---|
| Guo et al. 2021 [46] | Cross-sectional | US | >70 | 1053 (63) | Difficulty with walking from one room to another on the same level, getting in or out of bed, eating or drinking, and dressing oneself | Difficulty with doing chores around the house, preparing one's own meal, and managing one's money | Ophthalmologists Early Treatment Diabetic Retinopathy Study chart | Mild (20/40–20/60) Moderate or greater (<20–60) | Socio-demographic Health-related |
| Guthrie et al. 2018 [63] | Cross-sectional | Canada | >65 | 11829 (71) | Difficulty with getting across a room, dressing, bathing, eating, getting out of bed, or using the toilet | Difficulty with preparing a hot meal; shopping for groceries; making phone calls; taking medications; and managing money | Resident Assessment Instrument for Home Care and the Minimum Data Set 2.0 | NR | Socio-demographic Health-related |
| Harada et al. 2008 [64] | Cross-sectional | Japan | >65 | 843 (58) | NR | Tokyo Metropolitan Institute of Gerontology Index of Competence | Trained technicians | Corrected acuity of worse than 0.5 | Socio-demographic Health-related |
| Haymes et al. 2002 [51] | Cross-sectional | Australia | 20–89 | 120 (62) | Melbourne Low Vision ADL Index | Melbourne Low Vision IADL Index | Bailey–Lovie chart Pelli–Robson Chart | Retinitis pigmentosa (12.5%), Macular dystrophy (7.5%), Optic atrophy (5%), Diabetic retinopathy (3%), Glaucoma (2.5%), Myopic degeneration (2.5%), Retinal vein occlusion (1.5%), cataract (1.5%) | NR |
| Hochberg et al. 2012 [47] | Cross-sectional | US | 60–80 | 191 (58) | Katz Activities of Daily Living | Lawton instrumental activities of daily living | Ophthalmologists Early Treatment Diabetic Retinopathy Study chart | Glaucoma Age-related macular degeneration | Socio-demographic Health-related |
| Horowitz et al. 1994 [42] | Cohort | US | 44–99 | 114 (NR) | Monthly Nurse's Assessment | NR | Optometric Examination Record | Moderate: Corrected acuity 20/70-20/100 Severe: Corrected acuity ≥20/200 | Socio-demographic Health-related |
| Ivanoff et al. 2000 [54] | Cross-sectional | Sweden | 85 | 617 (35) | Katz Activities of Daily Living | Lawton instrumental activities of daily living | Monoyer chart | Cataract (54%), Cataract and other eye diseases (29%), Other eye diseases (17%) | NR |
| Kee et al. 2021 [11] | Cross-sectional | Malaysia | >60 | 208 (57) | NR | Lawton instrumental activities of daily living | Lighthouse near VA chart and Pelli-Robson CS chart | Near visual impairment | NR |
| Keller et al. 1999 [55] | Cross-sectional | US | >60 | 576 (72) | Katz Activities of Daily Living | Lawton instrumental activities of daily living | Lighthouse near VA chart | Corrected acuity ≥20/70 | Socio-demographic Health-related |

(*Continued*)

**Table 1.** (Continued)

| Study | Design | Country | Age (year) | Participant (Female%) | ADL measurement | IADL measurement | Vision assessment | Vision impairment characteristics | Adjusted variables |
|---|---|---|---|---|---|---|---|---|---|
| Laitinen et al. 2007 [50] | Cross-sectional | Finland | >55 | 2870 (60) | Katz Activities of Daily Living | Lawton instrumental activities of daily living | Snellen E Chart | Impaired: ≤0.25 (≤20/80) Reduced: 0.5–0.63 (20/40–20/32) Moderate: 0.32–0.4 (20/63–20/50) | Socio-demographic Health-related |
| Lam et al. 2013 [65] | Cohort | US | 65–84 | 2520 (58) | Katz Activities of Daily Living | Lawton instrumental activities of daily living | Early Treatment Diabetic Retinopathy Study chart | NR | Socio-demographic Health-related |
| Liu et al. 2016 [37] | Cohort | US | >65 | 3871 (65) | Katz Activities of Daily Living | Lawton instrumental activities of daily living | Self-reported vision impairment | NR | Socio-demographic Health-related |
| Mercan et al. 2021 [12] | Cross-sectional | Turkey | >65 | 578 (53) | NR | Lawton instrumental activities of daily living | International Classification of Functioning, Disability and Health | NR | Socio-demographic Health-related |
| Mueller-Schotte et al. 2019 [61] | Cohort | Netherland | >60 | 9319 (59) | NR | Modified KATZ-15 IADL questionnaire | International Classification of Functioning, Disability and Health | NR | Socio-demographic Health-related |
| Naël et al. 2017 [43] | Cross-sectional | France | >65 | 709 (65) | Katz Activities of Daily Living | Lawton instrumental activities of daily living | Ophthalmologists | > 20/32–20/25 > 20/40–20/32 > 20/63–20/40 ≤ 20/63 | Socio-demographic Health-related |
| Park et al. 2015 [59] | Cross-sectional | South Korea | >55 | 9047 (55) | Difficulty with dressing, washing face, bathing, feeding, transferring, using toilet, and incontinence | NR | Physician diagnosis | Glaucoma | Socio-demographic Health-related |
| Pér'es et al. 2017 [56] | Cohort Cross-sectional | France | >65 | 9294 (60) | Difficulty with bathing; dressing; eating; standing up from bed/chair or sitting down on a chair; walking indoors; and toileting | Difficulty with preparing own meals; shopping; managing money; using the telephone; doing housework; taking transportation; and taking prescribed medication | Parinaud scale | Corrected acuity < 20/30 | Socio-demographic Health-related |
| Qiu et al. 2014 [62] | Cross-sectional | US | >40 | 5186 (NR) | Katz Activities of Daily Living | Lawton instrumental activities of daily living | N-30–5 algorithm of FDT perimetry | NR | Socio-demographic Health-related |
| Reuben et al. 1999 [44] | Cohort | US | >60 | 5646 (53) | Katz Activities of Daily Living | Lawton instrumental activities of daily living | Ophthalmologists | Corrected acuity ≥20/40 | Socio-demographic Health-related |
| Rokicki et al. 2016 [45] | Cross-sectional | Poland | >55 | 623 (100) | NR | Lawton instrumental activities of daily living | Ophthalmic examination | Corrected acuity ≤ 0.7–0.3 | NR |

(*Continued*)

**Table 1.** (Continued)

| Study | Design | Country | Age (year) | Participant (Female%) | ADL measurement | IADL measurement | Vision assessment | Vision impairment characteristics | Adjusted variables |
|---|---|---|---|---|---|---|---|---|---|
| Ross et al. 1991 [52] | Cross-sectional | US | 33–94 | 144 (4) | NR | Identifying currency, reading a wristwatch, playing cards, using a ruler, dialing a telephone | Bailey-Lovie chart, Vistech VCTS 6500 charts | NR | NR |
| Rovner et al. 1998 [30] | Cohort | US | >68 | 872 (60) | NR | Older Americans Research and Service Center Instrument | Self-reported vision impairment | NR | Socio-demographic |
| Rubin et al. 1994 [57] | Cross-sectional | US | >65 | 222 (64) | Katz Activities of Daily Living | Lawton instrumental activities of daily living | Berkeley glare test, Randot circles test | NR | Socio-demographic Health-related |
| Swanson et al. 2004 [38] | Cohort | US | >18 | 67570 (54) | Katz Activities of Daily Living | Lawton instrumental activities of daily living | Self-reported vision impairment | NR | Socio-demographic Health-related |
| Tareque et al. 2019 [31] | Cohort | Singapore | >60 | 3452 (54) | Katz Activities of Daily Living | Lawton instrumental activities of daily living | Self-reported vision impairment | NR | Health-related |
| Verbeek et al. 2022 [13] | Cohort | Netherland | >85 | 548 (66) | Groningen Activity Restriction Scale | Groningen Activity Restriction Scale | Early Treatment Diabetic Retinopathy Study charts | Moderate (0.5≤ visual acuity ≤0.7) Severe visual impairment (visual acuity <0.5). | NR |
| Wahl et al. 1999 [58] | Cohort | Germany | >65 | 67 (71) | Schneekloth and Potthoff items | Schneekloth and Potthoff items | Visual acuity criterion of legal blindness in the United States | Visual acuity between 20/200 and 20/600 | Socio-demographic Health-related |
| Wallhagen et al. 2001 [39] | Cohort | US | >50 | 2442 (57) | Katz Activities of Daily Living | Lawton instrumental activities of daily living | Self-reported vision impairment | NR | Socio-demographic Health-related |
| West et al. 1997 [48] | Cohort | US | 65–84 | 4624 (59) | Katz Activities of Daily Living | Lawton instrumental activities of daily living | Ophthalmologists ETDRS charts Pelli Robson chart Humphrey 81-point full-field screen RANDOT circles test | Visual acuity worse than 20/40 | Socio-demographic Health-related |
| Whitson et al. 2007 [40] | Cohort | US | >65 | 3878 (64) | Katz Activities of Daily Living | Lawton instrumental activities of daily living | Self-reported vision impairment | NR | Socio-demographic Health-related |
| Zhang et al. 2022 [14] | Cohort | China | >80 | 1750 (72) | Barthel index | NR | Visual Analogue Scale | NR | Socio-demographic Health-related |

NR, Not reported.

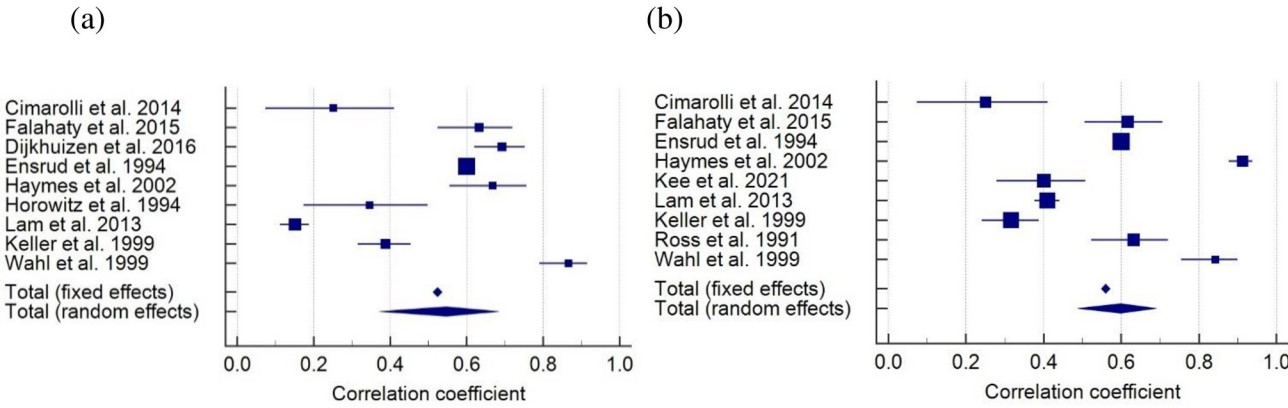

**Fig 2.** Forest plot of correlation between vision impairment and difficulties with (a) ADL and (b) IADL.

| Study or Subgroup | log[Odds Ratio] | SE | Weight | Odds Ratio IV, Random, 95% CI |
|---|---|---|---|---|
| **1.6.2 Study** | | | | |
| Bekibele et al. 2008 Distance vision | -0.211 | 0.244 | 2.2% | 0.81 [0.50, 1.31] |
| Bekibele et al. 2008 Near vision | 0.873 | 0.281 | 2.0% | 2.39 [1.38, 4.15] |
| Berger et al. 2008 Distance vision | 0.122 | 0.086 | 3.1% | 1.13 [0.95, 1.34] |
| Berger et al. 2008 Near vision | 0.565 | 0.071 | 3.1% | 1.76 [1.53, 2.02] |
| Brennan et al. 2005 | 0.412 | 0.066 | 3.2% | 1.51 [1.33, 1.72] |
| Cacciatore et al. 2004 | 0.802 | 0.153 | 2.8% | 2.23 [1.65, 3.01] |
| Cao et al. 2021 | 1.717 | 0.088 | 3.1% | 5.57 [4.69, 6.62] |
| Chan et al. 2021 Female | 0.978 | 0.343 | 1.7% | 2.66 [1.36, 5.21] |
| Chan et al. 2021 Male | 1.332 | 0.265 | 2.1% | 3.79 [2.25, 6.37] |
| Crews et al. 2004 | 1.019 | 0.059 | 3.2% | 2.77 [2.47, 3.11] |
| Ensrud et al. 1994 | 0.231 | 0.042 | 3.2% | 1.26 [1.16, 1.37] |
| Guo et al. 2021 Mild | 0.039 | 0.198 | 2.5% | 1.04 [0.71, 1.53] |
| Guo et al. 2021 Moderate to severe | 0.086 | 0.268 | 2.1% | 1.09 [0.64, 1.84] |
| Guthrie et al. 2018 | 0.399 | 0.026 | 3.3% | 1.49 [1.42, 1.57] |
| Ivanoff et al. 2000 | 0.3 | 0.221 | 2.4% | 1.35 [0.88, 2.08] |
| Laitinen et al. 2007 0.33-0.4 | 0.626 | 0.23 | 2.3% | 1.87 [1.19, 2.94] |
| Laitinen et al. 2007 0.5-0.63 | 0.392 | 0.174 | 2.6% | 1.48 [1.05, 2.08] |
| Latinen et al. 2007 ≤0.25 | 1.472 | 0.296 | 1.9% | 4.36 [2.44, 7.78] |
| Liu et al. 2016 | 0.688 | 0.187 | 2.6% | 1.99 [1.38, 2.87] |
| Nael et al. 2017 20/32-20/25 | -0.288 | 0.465 | 1.2% | 0.75 [0.30, 1.87] |
| Nael et al. 2017 20/40-20/32 | 0.3 | 0.355 | 1.7% | 1.35 [0.67, 2.71] |
| Nael et al. 2017 20/63 | 0.588 | 0.425 | 1.4% | 1.80 [0.78, 4.14] |
| Naet et al. 2017 20/63-20/40 | 0.425 | 0.371 | 1.6% | 1.53 [0.74, 3.16] |
| Park et al. 2015 | -0.062 | 0.446 | 1.3% | 0.94 [0.39, 2.25] |
| Peres et al. 2017 Distance & Near Vision (Cohort) | 0.642 | 0.288 | 2.0% | 1.90 [1.08, 3.34] |
| Peres et al. 2017 Distance & Near Vision (Cross) | 0.642 | 0.604 | 0.9% | 1.90 [0.58, 6.21] |
| Peres et al. 2017 Distance Vision (Cohort) | 0.405 | 0.32 | 1.8% | 1.50 [0.80, 2.81] |
| Peres et al. 2017 Distance Vision (Cross) | 0.642 | 0.526 | 1.0% | 1.90 [0.68, 5.33] |
| Peres et al. 2017 Near Vision (Cohort) | 0.47 | 0.188 | 2.6% | 1.60 [1.11, 2.31] |
| Peres et al. 2017 Near Vision (Cross) | 0.693 | 0.405 | 1.4% | 2.00 [0.90, 4.42] |
| Qiu et al. 2014 Mild | 0.148 | 0.225 | 2.3% | 1.16 [0.75, 1.80] |
| Qiu et al. 2014 Moderate | 0.27 | 0.237 | 2.3% | 1.31 [0.82, 2.08] |
| Qiu et al. 2014 Severe | 0.896 | 0.269 | 2.1% | 2.45 [1.45, 4.15] |
| Reuben et al. 1999 | 0.698 | 0.226 | 2.3% | 2.01 [1.29, 3.13] |
| Rubin et al. 1994 | 0.207 | 0.203 | 2.5% | 1.23 [0.83, 1.83] |
| Swanson et al. 2004 18-44 Years | 1.346 | 0.138 | 2.8% | 3.84 [2.93, 5.04] |
| Swanson et al. 2004 45-64 Years | 0.443 | 0.093 | 3.1% | 1.56 [1.30, 1.87] |
| Swanson et al. 2004 ≥65 Years | 0.261 | 0.061 | 3.2% | 1.30 [1.15, 1.46] |
| Tareque et al. 2019 | 1.033 | 0.139 | 2.8% | 2.81 [2.14, 3.69] |
| Wallhagen et al. 2001 Mild | 0.405 | 0.172 | 2.7% | 1.50 [1.07, 2.10] |
| Wallhagen et al. 2001 Moderate to severe | 0.829 | 0.219 | 2.4% | 2.29 [1.49, 3.52] |
| West et al. 1997 | 0.599 | 0.223 | 2.4% | 1.82 [1.18, 2.82] |
| Whitson et al. 2007 | 0.519 | 0.149 | 2.8% | 1.68 [1.25, 2.25] |
| **Subtotal (95% CI)** | | | **100.0%** | **1.77 [1.56, 2.01]** |

Heterogeneity: Tau² = 0.13; Chi² = 477.98, df = 42 (P < 0.00001); I² = 91%
Test for overall effect: Z = 8.77 (P < 0.00001)

| | | | | |
|---|---|---|---|---|
| **Total (95% CI)** | | | **100.0%** | **1.77 [1.56, 2.01]** |

Heterogeneity: Tau² = 0.13; Chi² = 477.98, df = 42 (P < 0.00001); I² = 91%
Test for overall effect: Z = 8.77 (P < 0.00001)
Test for subgroup differences: Not applicable

**Fig 3. Forest plot of the association between vision impairment and difficulties with activity of daily living.**

(a)

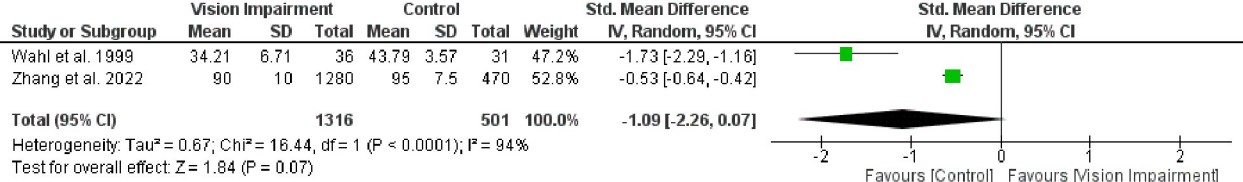

(b)

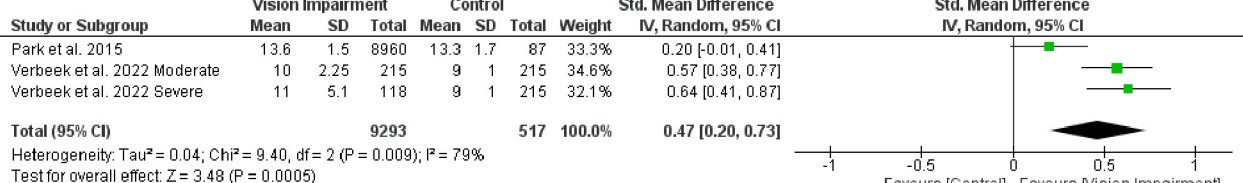

**Fig 4. Forest plot of the association between vision impairment and difficulties with activity of daily living using two different pooled SMD analysis.**

### 3.2. Correlation between vision impairment and difficulties with activity of daily living

Eight studies [9, 10, 36, 42, 51, 53, 55, 58, 65] involving 10,700 participants reported a correlation between vision impairment and difficulties with ADL. The pooled correlation coefficient was 0.55 (95% CI 0.37–0.68, $p = 0.001$), indicating a positive large correlation between vision impairment and difficulties with ADL (Fig 2A). The values of $I^2 = 99\%$ ($p < 0.0001$) indicated that significant heterogeneity exists in the included studies. Additionally, nine studies [10, 11, 36, 51–53, 55, 58, 65] involving 11,088 participants reported a correlation between vision impairment and difficulties with IADL. The pooled correlation coefficient showed a positive large correlation between vision impairment and difficulties with IADL ($r = 0.60$, 95% CI 0.49–0.69, $p = 0.001$) (Fig 2B). There was evidence of significant heterogeneity across included studies ($I^2 = 97\%$, $p < 0.0001$).

### 3.3. Association of vision impairment and difficulties with activities of daily living

Twenty-three studies [7, 8, 28, 31, 32, 34, 35, 37–39, 43, 44, 46, 48, 50, 53, 54, 56, 57, 59, 60, 62, 63] involving 182,743 participants reported association between vision impairment and difficulties with ADL. The random-effect model by pooling log- odds ratio using generic inverse variance meta-analysis showed that vision impairment was significantly associated with difficulties in activity of daily living (OR = 1.77, 95% CI 1.56–2.01, $p < 0.0001$) (Fig 3). The values of $I^2 = 91\%$ ($p < 0.0001$) indicated that significant heterogeneity exists in the included studies. Additionally, the pooled SMD analyses from Wahl et al. 1999 [58] and Zhang et al. 2022 [14] studies indicated a trend toward association between vision impairment and difficulties with ADL (SMD = -1.09, 95% CI -2.26–0.07, $p = 0.07$) (Fig 4A). Moreover, using a different assessment criterion from Park et al. 2015 [59] and Verbeek et al. 2022 [13] studies indicated significant association between vision impairment and difficulties with ADL (SMD = 0.47, 95% CI 0.20–0.73, $p = 0.0005$) (Fig 4B).

Stratifying studies by different assessment of vision impairment showed non-significant higher difficulties in studies that used self-reported assessment (OR = 1.80, 95% CI 1.55–2.09,

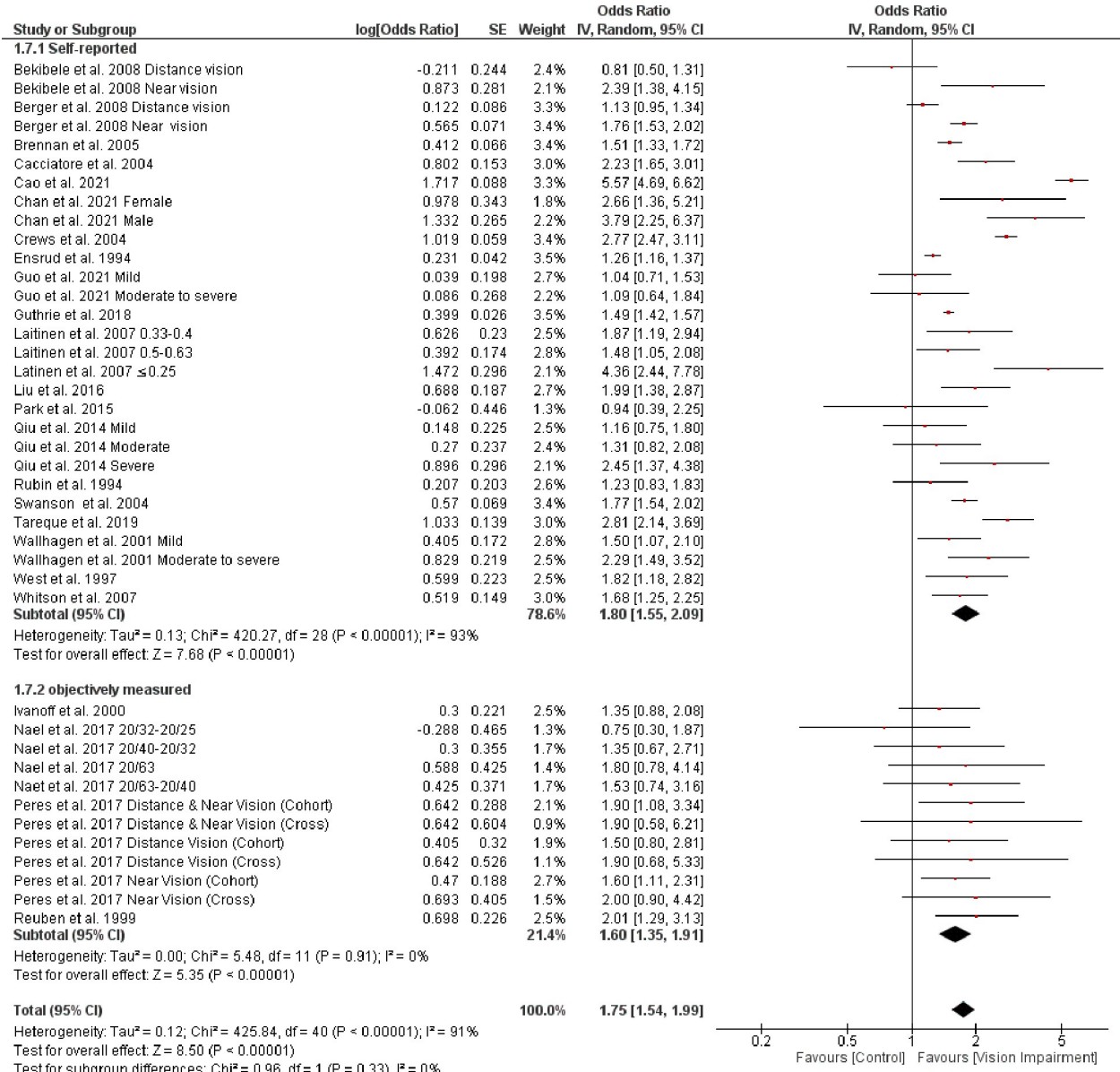

**Fig 5. Forest plot of the association between vision impairment and difficulties with activity of daily living based on different ADL assessments.**

$p < 0.0001$, $I^2 = 93\%$, $p < 0.0001$) compared with studies that objectively assessed vision impairment (OR = 1.60, 95% CI 1.35–1.91, $p < 0.0001$, $I^2 = 0\%$, $p = 0.91$) (test for subgroup difference: Chi$^2$ = 0.96, $p = 0.33$) (Fig 5). However, the result of sensitivity analysis indicated that after removing Cao et al. 2021 [60], the heterogeneity dropped to 46% (Fig 6).

Subgroup analysis based on severity of vision impairment revealed higher ADL difficulties with moderate to severe impairment (visual acuity between 20/70 to 20/160 and worse than 20/200 in the better seeing eye) (OR = 1.78, 95% CI 1.43–2.21, $p = 0.00001$, $I^2 = 44\%$, $p = 0.02$) compared with mild to moderate impairment (visual acuity $\leq$ 20/200-20/70 in the better seeing eye) (OR = 1.28, 95% CI 1.06–2.54, $p = 0.5$, $I^2 = 38\%$, $p = 0.01$) (test for subgroup difference: Chi$^2$ = 4.95, $p = 0.03$) (Fig 7).

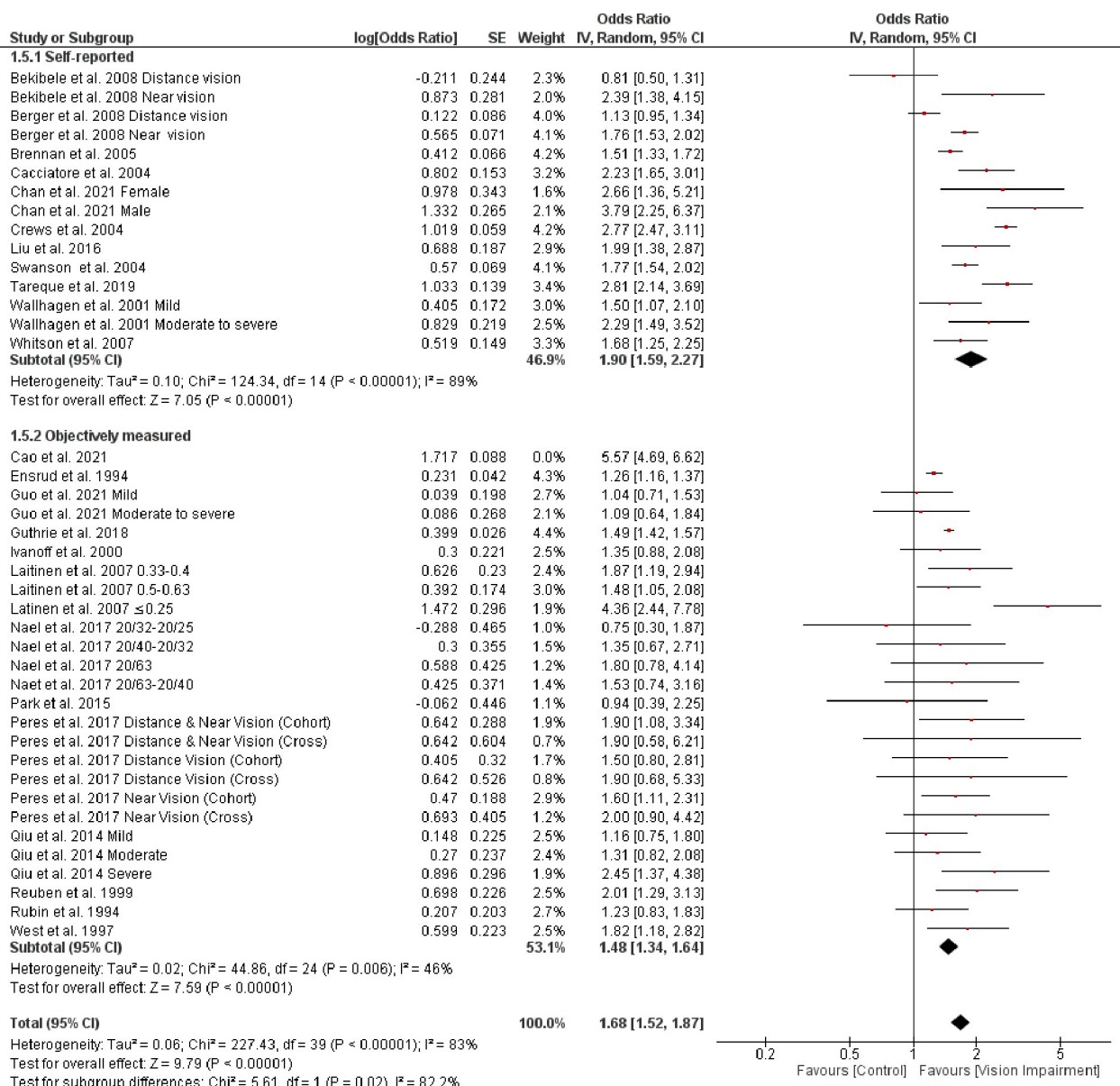

**Fig 6. Sensitivity analysis for the association between vision impairment and difficulties with activity of daily living based on different ADL assessments.**

Further, subgroup analysis based on different assessment of ADL showed a non-significant higher difficulty in studies that used self-reported assessment (OR = 1.80, 95% CI 1.55–2.09, $p < 0.0001$, $I^2 = 93\%$, $p < 0.0001$) compared with studies that objectively assessed ADL (OR = 1.60, 95% CI 1.35–1.91, $p < 0.0001$, $I^2 = 0\%$, $p = 0.91$) (test for subgroup difference: Chi$^2$ = 0.96, $p = 0.33$) (Fig 8). Although, the heterogeneity dropped to zero in studies that objectively assessed ADL, implicating that the source of heterogeneity is related to the assessment method of ADL.

Finally, subgroup analysis based on different vision impairment characteristics showed significant higher difficulty in ADL in people with distance vision impairment (OR = 1.12, 95% CI 0.90–1.40, $p = 0.30$, $I^2 = 17\%$, $p = 0.31$) and also in people with both distance and near vision

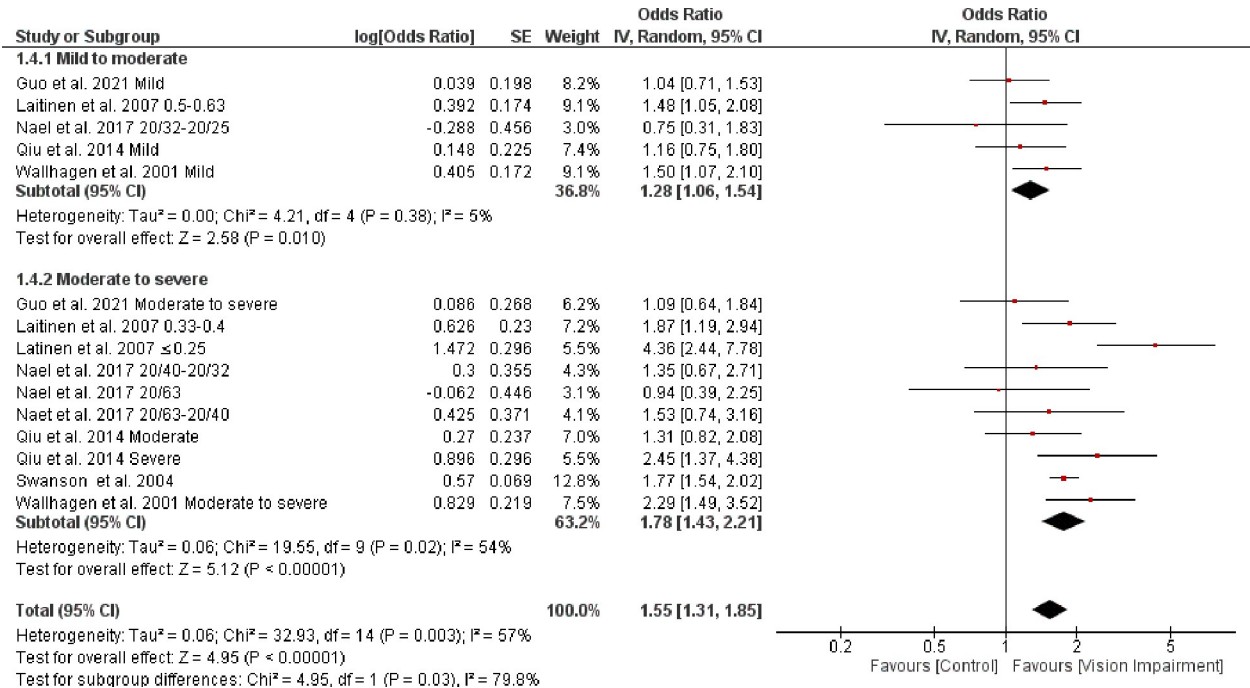

**Fig 7. Forest plot of the association between vision impairment and difficulties with activity of daily living based on severity of vision impairment.**

impairments (OR = 1.90, 95% CI 1.14–3.16, $p$ = 0.01, $I^2$ = 0%, $p$ = 1.00) compared with people with near vision impairment (OR = 1.77, 95% CI 1.57–2.01, $p$ < 0.0001, $I^2$ = 0%, $p$ = 0.67) (test for subgroup difference: Chi$^2$ = 12.83, $p$ = 0.002) (Fig 9). Importantly, heterogeneity dropped to zero and 17% for these three analyses showing that vision impairment characteristics was another source of heterogeneity.

## 3.4. Association of vision impairment and difficulties with Instrumental activities of daily living (IADL)

Thirty-two studies [7, 11, 12, 28–35, 37–41, 43, 44, 46–50, 53, 54, 56, 57, 60–64] involving 193,485 participants reported associations between vision impairment and difficulties with IADL. Overall pooled analyses showed that there is a significant association between vision impairment and difficulties with IADL (OR = 1.96, 95% CI 1.68–2.30, $p$ < 0.0001, $I^2$ = 96%, $p$ < 0.0001) (Fig 10).

Subgroup analysis based on the different assessment of vision impairment showed higher odds of IADL in studies that used self-reported assessment (OR = 2.19, 95% CI 1.85–2.61, $p$ = 0.00001, $I^2$ = 89%, $p$ < 0.0001) compared with studies that objectively assessed vision impairment (OR = 1.83, 95% CI 1.46–2.28, $p$ < 0.0001, $I^2$ = 97%, $p$ < 0.0001). However, the test for subgroup difference was not statistically significant (Chi$^2$ = 1.65, $p$ = 0.20) (Fig 11). Further, subgroup analysis based on severity of vision impairment revealed a higher significant difficulty in IADL in participants with moderate to severe vision impairment (visual acuity between 20/70 to 20/160 and worse than 20/200 in the better seeing eye)) (OR = 1.86, 95% CI 1.57–2.20, $p$ = 0.00001, $I^2$ = 56%, $p$ = 0.007) compared with participants with mild to moderate vision impairment (visual acuity ≤ 20/200-20/70 in the better seeing eye)) (OR = 1.38, 95% CI 1.23–1.55, $p$ < 0.0001, $I^2$ = 0%, $p$ = 0.50) (test for subgroup difference: Chi$^2$ = 8.21, $p$ = 0.004)

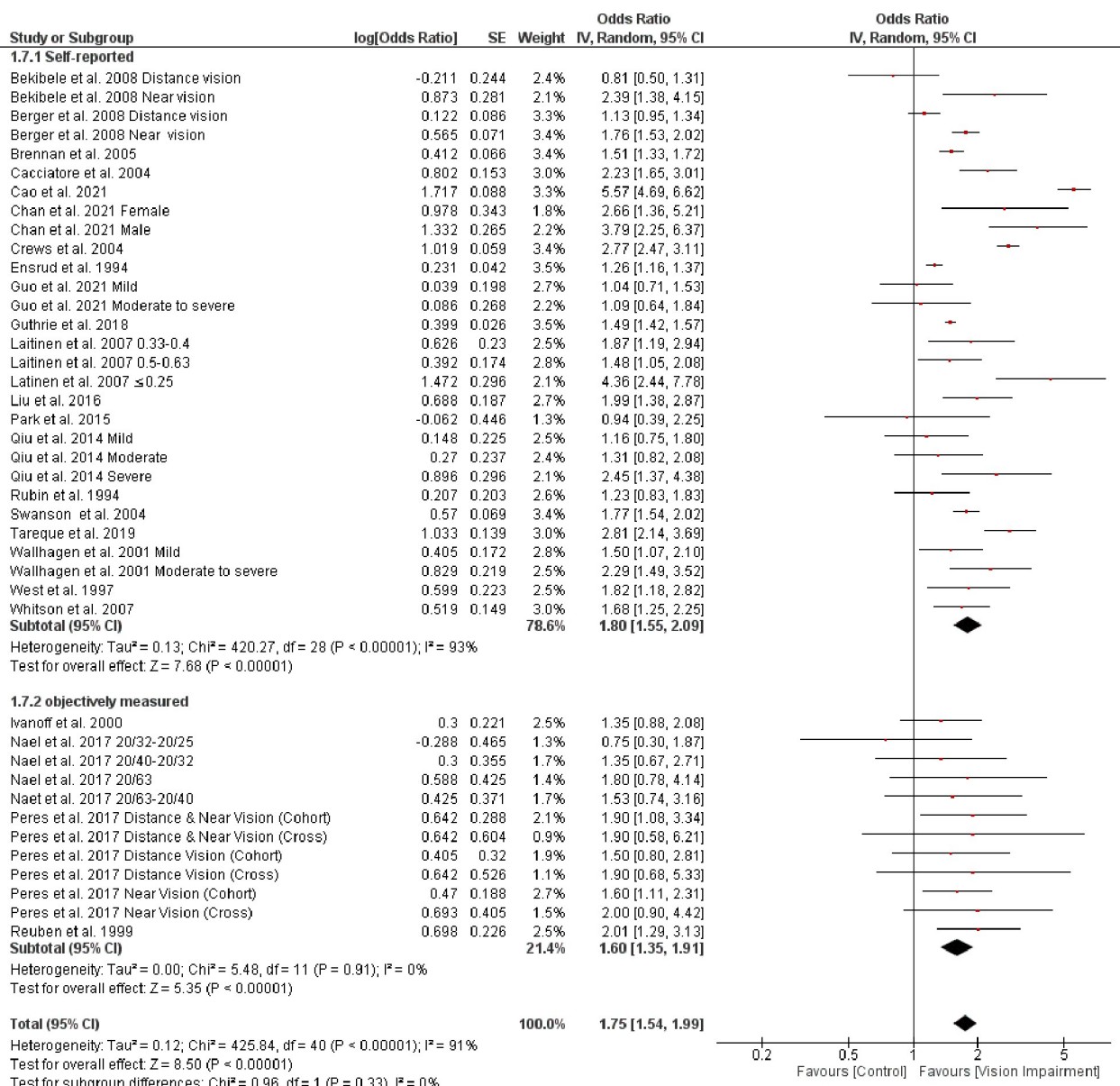

**Fig 8. Forest plot of the association between vision impairment and difficulties with activity of daily living based on different assessments of ADL.**

(Fig 12). Further, subgroup analysis based on the different assessments of IADL showed a significant trend toward higher difficulty in IADL in studies that used self-reported assessment (OR = 2.07, 95% CI 1.72–2.49, $p < 0.0001$, $I^2 = 97\%$, $p < 0.0001$) compared with studies that objectively assessed IADL (OR = 1.60, 95% CI 1.28–2.00, $p < 0.0001$, $I^2 = 81\%$, $p < 0.0001$) (test for subgroup difference: Chi$^2$ = 3.05, $p = 0.08$) (Fig 13). The result of sensitivity analysis revealed that after removing data for distance and near vision impairment from the cross-sectional study Peres et al. 2017 [56], the heterogeneity dropped to zero, implicating that the source of heterogeneity is related to the assessment method of IADL (Fig 14). Finally, subgroup analysis based on different vision impairment characteristics showed significant higher difficulty in ADL in people with near vision impairment (OR = 1.79, 95% CI 1.32–2.42,

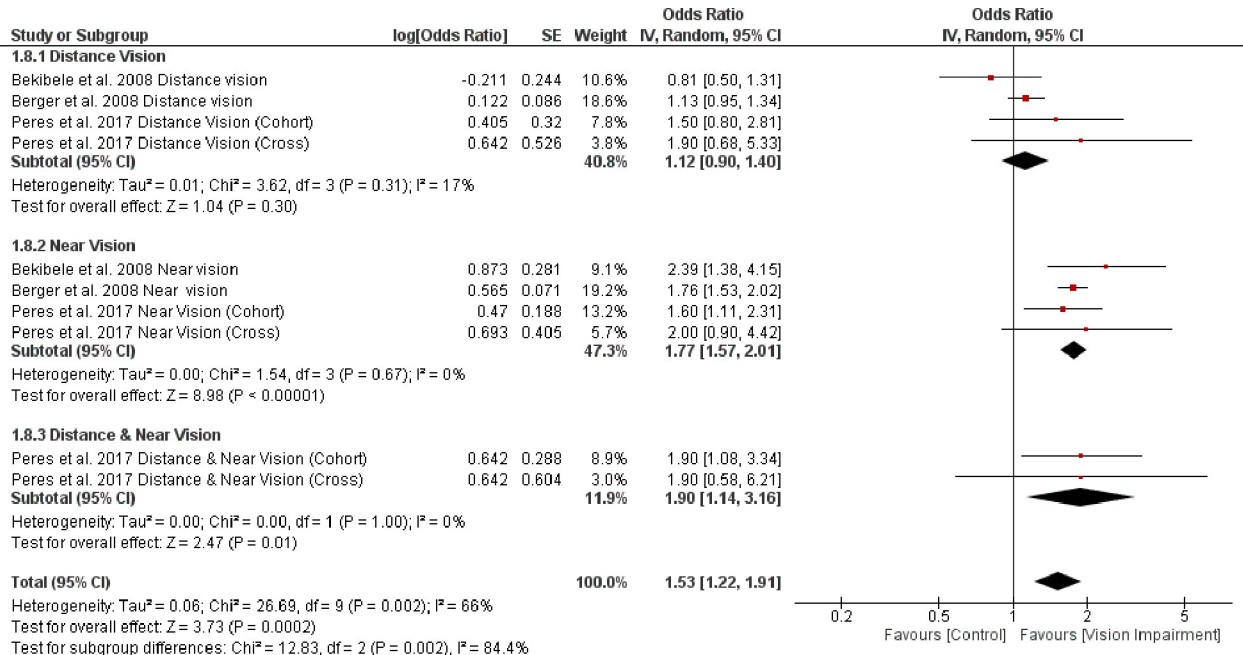

**Fig 9. Forest plot of the association between vision impairment and difficulties with activity of daily living based on different vision impairment characteristics.**

$p = 0.0002$, $I^2 = 87\%$, $p < 0.0001$) compared with people with distance vision impairment (OR = 1.19, 95% CI 1.05–1.34, $p = 0.005$, $I^2 = 0\%$, $p = 0.65$) and also in people with both distance and near vision impairments (OR = 3.19, 95% CI 0.93–10.96, $p = 0.07$, $I^2 = 96\%$, $p < 0.0001$) (test for subgroup difference: Chi$^2$ = 8.14, $p = 0.02$) (Fig 15).

### 3.5. Meta-regression analysis

To explore the potential sources of heterogeneity and examine the moderating role of age on the relationship between vision impairment and the performance of individuals in ADL and IADL, meta-regression analysis was performed. Age was used as the primary moderator variable in the regression model. The analysis revealed a significant negative association between age and both ADL and IADL performance. For ADL, the slope was -0.0147 (95% CI: -0.0179 to -0.0116, $p < 0.001$; S2 Fig), indicating that for every one-year increase in age, ADL performance decreased by an average of 0.0147 units. For IADL, the slope was -0.0047 (95% CI: -0.0088 to -0.0005, p = 0.031; S3 Fig), suggesting that IADL performance also declined with age, though the effect size was smaller compared to ADL. These findings highlight the significant moderating impact of age on functional performance, particularly in individuals with vision impairment.

## 4. Discussion

Our study shows strong evidence that vision impairment is associated and correlated with higher difficulties in ADL and IADL. The results of the present systematic review and meta-analysis also revealed that the poorer the vision impairment, the more severe the ADL and IADL disability exists in adults and older adults. We found that the association between vision impairment and difficulties in IADL is higher in studies with self-reported vision assessment compared with studies that objectively assessed vision impairment. We also found that the

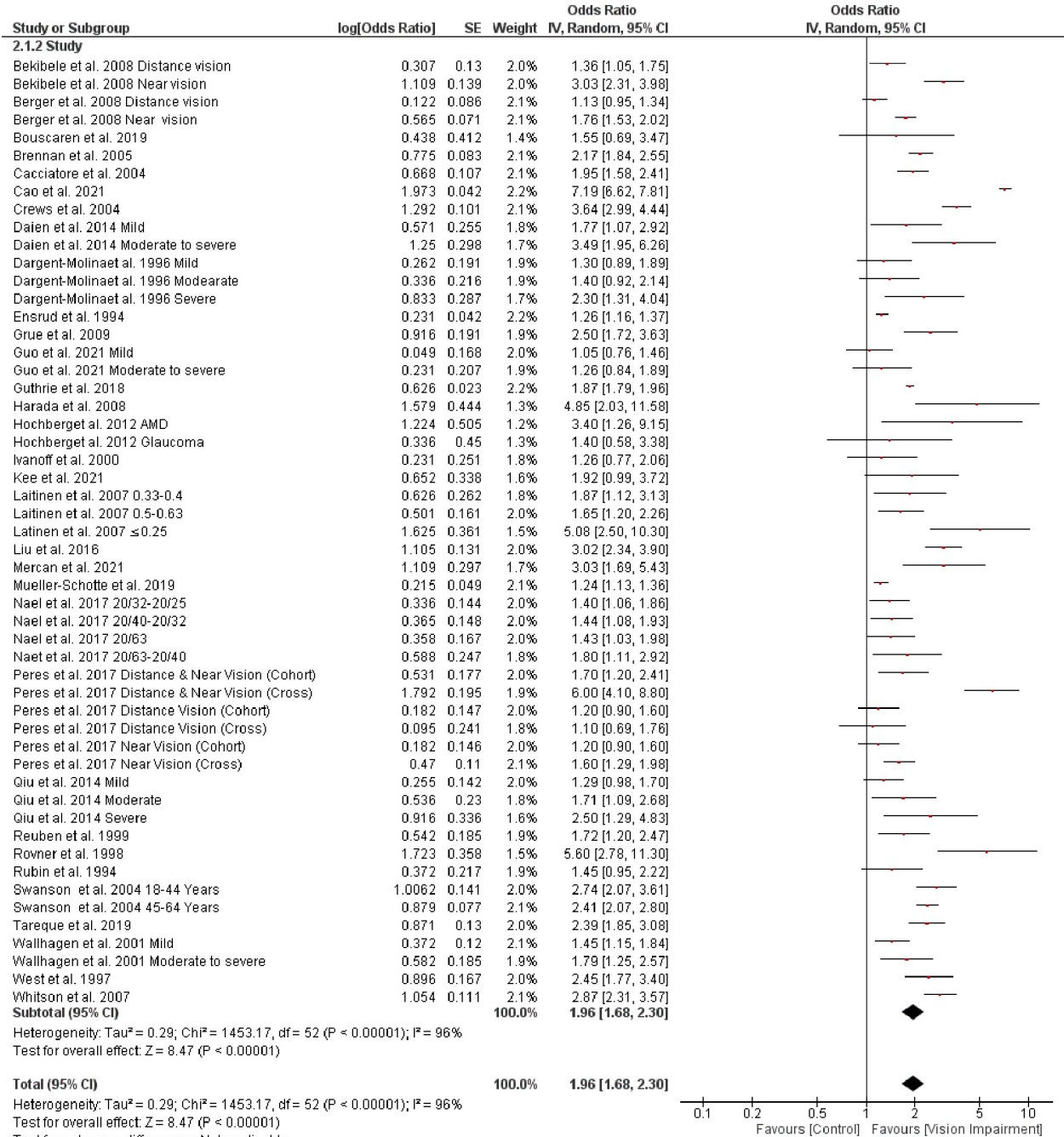

**Fig 10. Forest plot of the association between vision impairment and difficulties with instrumental activity of daily living.**

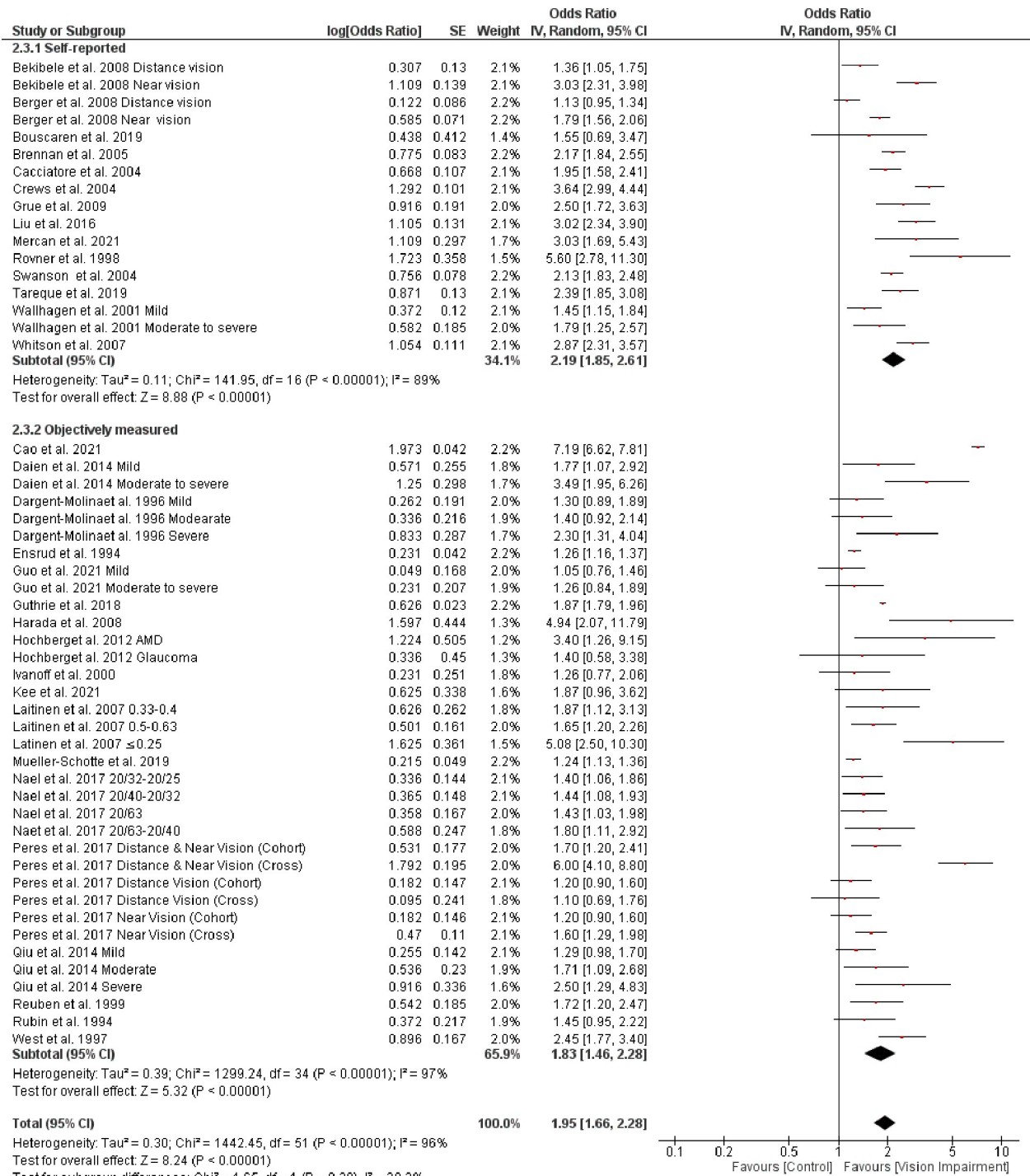

**Fig 11. Forest plot of the association between vision impairment and difficulties with instrumental activity of daily living based on different vision assessments.**

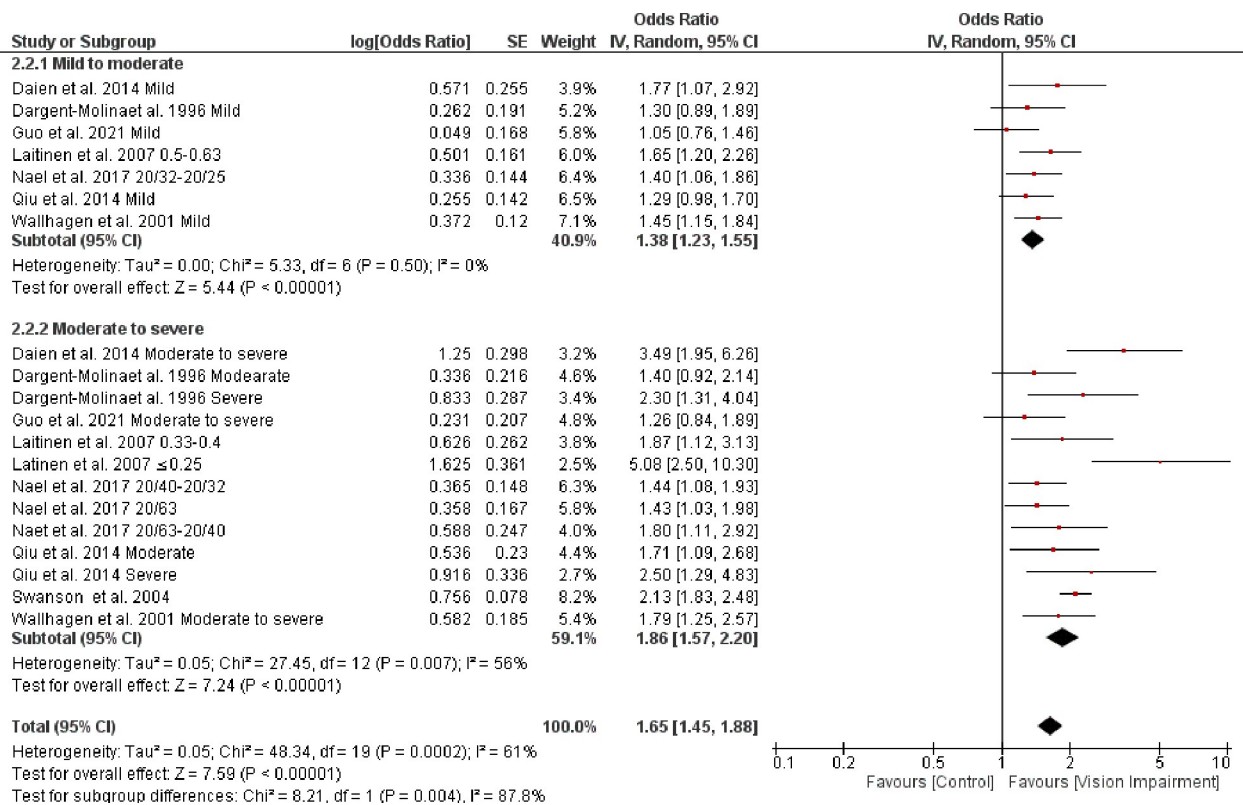

**Fig 12. Forest plot of the association between vision impairment and difficulties with instrumental activity of daily living based on severity of vision assessments.**

association between vision impairment and difficulties in ADL and IADL is higher in people with near vision impairment compared with people with only distance vision impairment.

Vision plays an important role in performing activities of daily living by continuously providing information about environment, and body movement and position to the nervous system [66]. People with vision impairment experience difficulties in carrying out reading, leisure activities, and activities of daily living [12, 43, 51, 56, 64, 67]. Vision impairment leads to reduction in the proprioception and vestibular inputs that are required for leisure activities, and activities of daily living [68, 69]. Therefore, it is important to ensure that vision impairment in adults and older adults is adequately treated or corrected, especially among those with ADL and IADL difficulties, in order to limit limitations of vision impairment on their lives. Among included studies in the present meta-analysis, only Park et al. 2015 analyzed the individual components of ADL and found significant higher prevalence of the bathing dimension than the healthy participants [59]. More studies are warranted to demonstrate which components of ADL and IADL are more affected from vision impairment.

Vision impairment affects quality of life and increases the risk of death, thus those with vision impairment require promotional, preventive, treatment, and rehabilitative interventions [1, 70]. The majority of individuals with vision impairment have some useful residual vision and would benefit from low-vision rehabilitation eye care health services [67]. Vision rehabilitation services involves the provision of devices to enhance residual vision, and devices or training techniques for performing tasks and daily activities without reliance on vision [67, 71]. Accordingly, The Lancet Global Health Commission on Global Eye Health emphasizes the importance of integrating prevention, treatment, and rehabilitation services for various eye

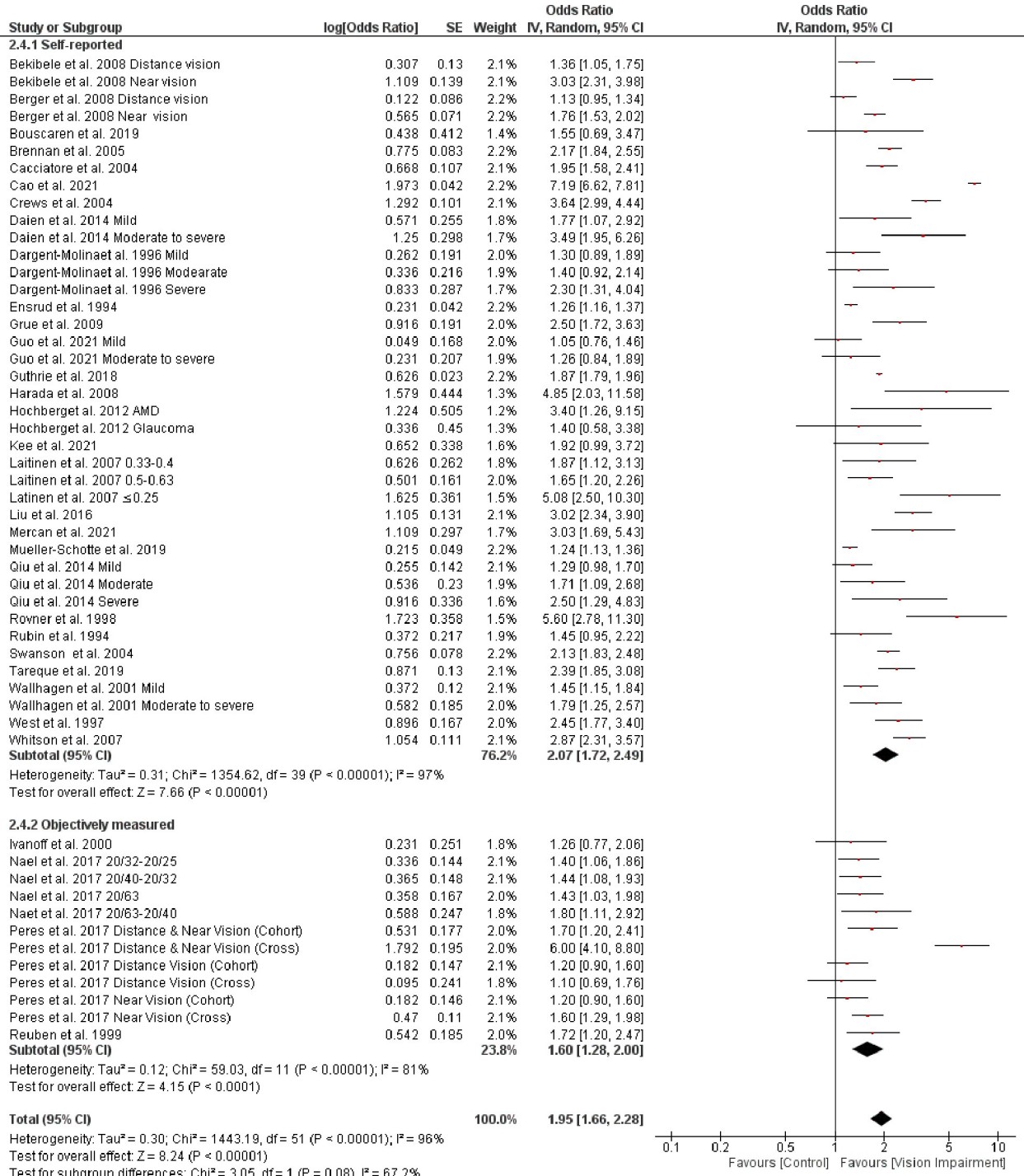

**Fig 13. Forest plot of the association between vision impairment and difficulties with instrumental activity of daily living based on different assessments of IADL.**

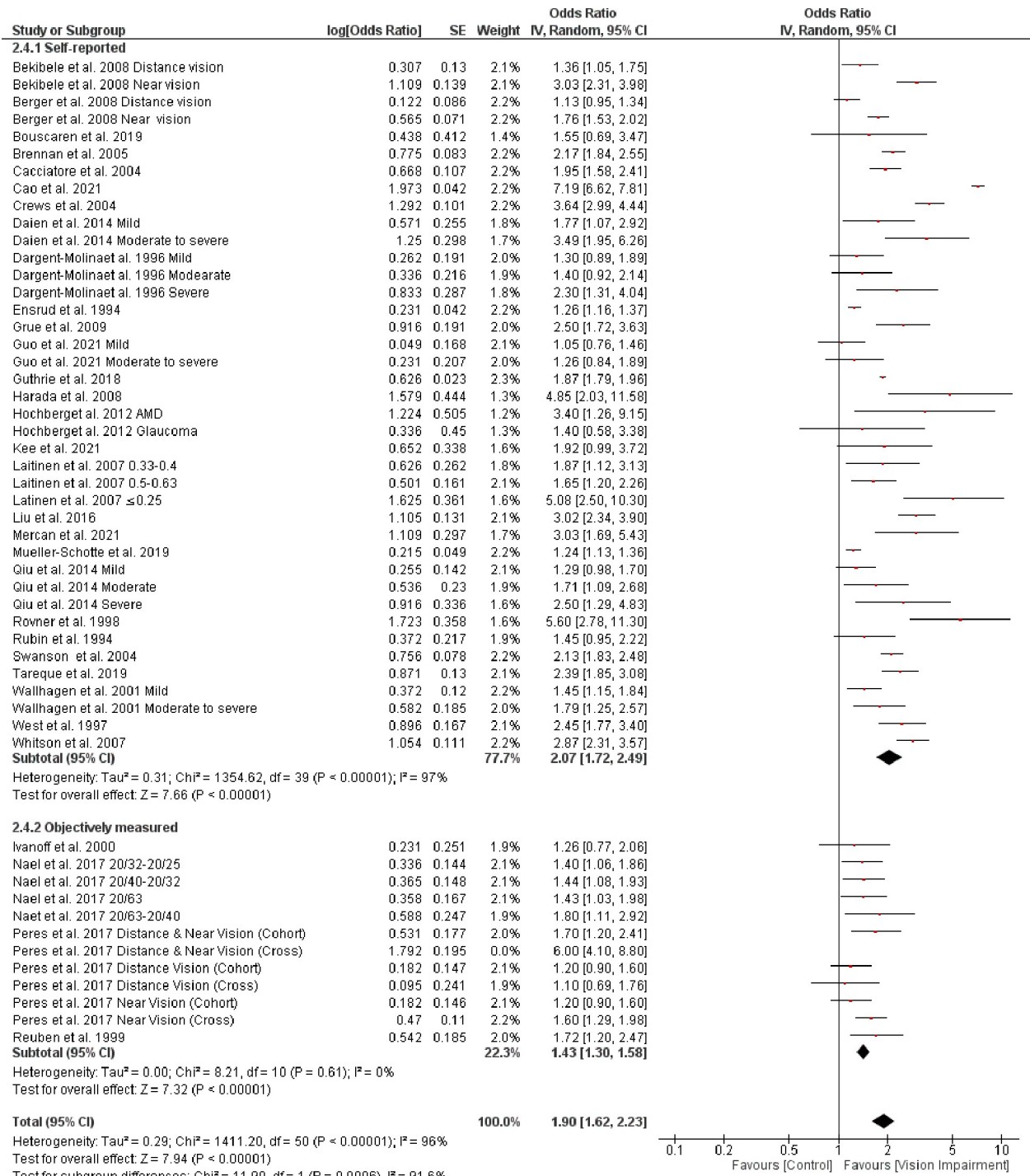

**Fig 14. Sensitivity analysis for the association between vision impairment and difficulties with instrumental activity of daily living based on different vision impairment characteristics.**

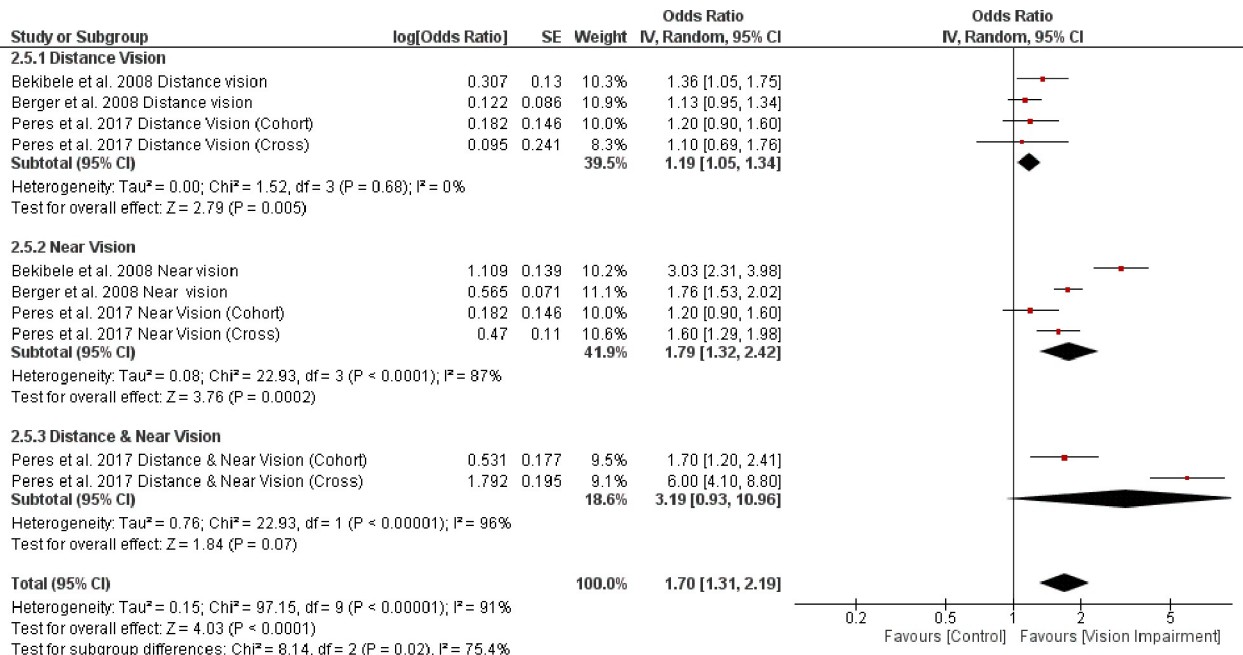

**Fig 15. Forest plot of the association between vision impairment and difficulties with instrumental activity of daily living based on different vision impairment characteristics.**

conditions into national health strategies, aligning them with the principles of universal health coverage [3]. Vision rehabilitation centers play a critical role in supporting individuals with vision disabilities by enabling them to attain and maintain independence and optimal functionality [3, 72]. These centers should prioritize services that enhance daily living activities, prevent accidents, and promote overall physical and mental wellbeing [3].

This systematic review and meta-analysis has a number of limitations. First, significant statistical heterogeneity was observed in the results, which can be attributed to variations in patient characteristics, vision assessment methods and definition, and the different ADL and IADL assessment tools used across studies. This heterogeneity arises not only from measurement error but also from the inherent differences in study designs, as detailed in Table 1. While we addressed this issue through subgroup analysis based on the different ADL and IADL measures, we have further discussed these differences to clarify the sources of heterogeneity and their potential impact on the findings. Second, different types of visual acuity charts were used in included studies to assess the associations between vision impairment and ADL and IADL difficulties. Third, extracted data on the difficulties in ADL and IADL in most of the included studies were based on self-reported information. Future studies should consider utilizing objective assessment of ADL and IADL by a trained neuropsychologist, occupational therapist, or health-related expert.

The results of the current systematic review and meta-analysis by using several statistical methods indicates that vision impairment is significantly associated with difficulties in functioning in a wide range of everyday activities, even for a minimal vision impairment level. This suggests that vision impairment is a predictive factor for accelerated deterioration in physical functioning, mainly for activities in daily living. The current systematic review and meta-analysis indicates that vision impairment remains an urgent and increasingly important public health priority.

## Supporting information

**S1 Checklist. PRISMA 2020 checklist.**
(DOCX)

**S2 Checklist. MOOSE checklist.**
(DOCX)

**S1 Table. Literature search strategy.**
(DOCX)

**S2 Table. Characteristics of individual studies.**
(XLSX)

**S3 Table. A list of the excluded studies and reasons for their exclusion.**
(DOCX)

**S4 Table. Quality assessment and publication bias evaluation of included study using the Newcastle-Ottawa Scale (NOS).**
(DOCX)

**S1 Fig. Funnel plot for publication bias.**
(DOCX)

**S2 Fig. Meta-regression analysis for the association between vision impairment and difficulties with activity of daily living based on age.**
(DOCX)

**S3 Fig. Meta-regression analysis for the association between vision impairment and difficulties with instrumental activity of daily living based on age.**
(DOCX)

## Author Contributions

**Conceptualization:** Masoud Rahmati, Mapa Prabhath Piyasena, Shahina Pardhan.

**Data curation:** Masoud Rahmati, Dong Keon Yon, Mapa Prabhath Piyasena, Shahina Pardhan.

**Formal analysis:** Masoud Rahmati, Dong Keon Yon, Hayeon Lee, Mapa Prabhath Piyasena, Shahina Pardhan.

**Funding acquisition:** Shahina Pardhan.

**Investigation:** Masoud Rahmati, Lee Smith, Laurent Boyer, Guillaume Fond, Dong Keon Yon, Hayeon Lee, Tarnjit Sehmbi, Mapa Prabhath Piyasena, Shahina Pardhan.

**Methodology:** Masoud Rahmati, Lee Smith, Laurent Boyer, Guillaume Fond, Dong Keon Yon, Hayeon Lee, Tarnjit Sehmbi, Shahina Pardhan.

**Project administration:** Masoud Rahmati, Lee Smith, Shahina Pardhan.

**Resources:** Masoud Rahmati, Shahina Pardhan.

**Software:** Guillaume Fond, Dong Keon Yon, Tarnjit Sehmbi, Shahina Pardhan.

**Supervision:** Masoud Rahmati, Lee Smith, Laurent Boyer, Mapa Prabhath Piyasena, Shahina Pardhan.

**Validation:** Masoud Rahmati, Laurent Boyer, Guillaume Fond, Hayeon Lee.

**Visualization:** Mapa Prabhath Piyasena, Shahina Pardhan.

**Writing – original draft:** Masoud Rahmati, Lee Smith, Laurent Boyer, Guillaume Fond, Dong Keon Yon, Hayeon Lee, Tarnjit Sehmbi, Mapa Prabhath Piyasena, Shahina Pardhan.

**Writing – review & editing:** Masoud Rahmati, Lee Smith, Laurent Boyer, Guillaume Fond, Dong Keon Yon, Hayeon Lee, Tarnjit Sehmbi, Mapa Prabhath Piyasena, Shahina Pardhan.

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
