## [Decision Letter · Decision Letter 0]

6 Nov 2024

PONE-D-24-38439Vision impairment and associated daily activity limitation: A systematic review and meta-analysisPLOS ONE

Dear Dr. Rahmati,

Thank you for submitting your manuscript to PLOS ONE. After careful consideration, we feel that it has merit but does not fully meet PLOS ONE’s publication criteria as it currently stands. Therefore, we invite you to submit a revised version of the manuscript that addresses the points raised during the review process.

Dear Dr Rahmati,

Thank you for your submission to PLOS ONE. Again, I apologize for the delay in this review. As discussed in our previous email correspondence, it was challenging to find reviewers during this busy end of year time. However, we've received decisions from two high caliber reviewers, and I am pleased to inform you that I am happy to accept your manuscript pending what I would term minor revisions. In addition to addressing Reviewer 1's relatively minor comments, **I would like you to consider including additional subgroup analyses stratified by age as Reviewer 2 suggests.** I agree that age is an important moderator of both vision impairment and worse IADL. A pooled analysis, and an additional analysis of only 65+, may be washing out some age effects. If you choose not to include these additional analyses, I would like to see a strong theoretical justification as to why. Additionally, please address Reviewer 2's minor comments as well. Thanks in advance,

Katya Numbers

We look forward to receiving your revised manuscript.

Kind regards,

Katya Numbers, PhD, M.S., B.S.

Academic Editor

PLOS ONE

Journal requirements: When submitting your revision, we need you to address these additional requirements. 1. Please ensure that your manuscript meets PLOS ONE's style requirements, including those for file naming. The PLOS ONE style templates can be found at https://journals.plos.org/plosone/s/file?id=wjVg/PLOSOne_formatting_sample_main_body.pdf and https://journals.plos.org/plosone/s/file?id=ba62/PLOSOne_formatting_sample_title_authors_affiliations.pdf 2. Please match your authorship list in your manuscript file and in the system. 3. As required by our policy on Data Availability, please ensure your manuscript or supplementary information includes the following:  A numbered table of all studies identified in the literature search, including those that were excluded from the analyses.   For every excluded study, the table should list the reason(s) for exclusion.   If any of the included studies are unpublished, include a link (URL) to the primary source or detailed information about how the content can be accessed.  A table of all data extracted from the primary research sources for the systematic review and/or meta-analysis. The table must include the following information for each study:  Name of data extractors and date of data extraction  Confirmation that the study was eligible to be included in the review.   All data extracted from each study for the reported systematic review and/or meta-analysis that would be needed to replicate your analyses.  If data or supporting information were obtained from another source (e.g. correspondence with the author of the original research article), please provide the source of data and dates on which the data/information were obtained by your research group.  If applicable for your analysis, a table showing the completed risk of bias and quality/certainty assessments for each study or outcome.  Please ensure this is provided for each domain or parameter assessed. For example, if you used the Cochrane risk-of-bias tool for randomized trials, provide answers to each of the signalling questions for each study. If you used GRADE to assess certainty of evidence, provide judgements about each of the quality of evidence factor. This should be provided for each outcome.   An explanation of how missing data were handled.   This information can be included in the main text, supplementary information, or relevant data repository. Please note that providing these underlying data is a requirement for publication in this journal, and if these data are not provided your manuscript might be rejected.   4. We noticed you have some minor occurrence of overlapping text with the following previous publication(s), which needs to be addressed: https://bmcresnotes.biomedcentral.com/articles/10.1186/s13104-021-05813-3https://www.ajo.com/article/S0002-9394(24)00335-0/fulltexthttps://onlinelibrary.wiley.com/doi/10.1002/jmv.28833https://researchonline.lshtm.ac.uk/id/eprint/4659682/1/PIIS2214109X20304885.pdf In your revision ensure you cite all your sources (including your own works), and quote or rephrase any duplicated text outside the methods section. Further consideration is dependent on these concerns being addressed. 5. Thank you for stating the following financial disclosure:  [This study was supported by Vision and Eye Research Institute, School of Medicine, Anglia Ruskin University, Young Street, Cambridge, United Kingdom.].  Please state what role the funders took in the study.  If the funders had no role, please state: ""The funders had no role in study design, data collection and analysis, decision to publish, or preparation of the manuscript."" If this statement is not correct you must amend it as needed. Please include this amended Role of Funder statement in your cover letter; we will change the online submission form on your behalf. 6. Please amend the manuscript submission data (via Edit Submission) to include author Dong Keon Yon. 7. Please include captions for your Supporting Information files at the end of your manuscript, and update any in-text citations to match accordingly. Please see our Supporting Information guidelines for more information: http://journals.plos.org/plosone/s/supporting-information. 

Reviewers' comments:

Reviewer's Responses to Questions

**Comments to the Author**

1. Is the manuscript technically sound, and do the data support the conclusions?

Reviewer #1: Yes

Reviewer #2: Yes

2. Has the statistical analysis been performed appropriately and rigorously? 

Reviewer #1: Yes

Reviewer #2: No

3. Have the authors made all data underlying the findings in their manuscript fully available?

Reviewer #1: Yes

Reviewer #2: Yes

4. Is the manuscript presented in an intelligible fashion and written in standard English?

Reviewer #1: Yes

Reviewer #2: Yes

5. Review Comments to the Author

Reviewer #1: PONE-D-24-38439_Review

Vision impairment and associated daily activity limitation: A systematic review 2 and meta-analysis

Generally a well thought, well written manuscript.

Abstract

Background: Doesn’t reflect scholarly inquest or bring out the problem that the researchers sought to study.

Review the grammar of sentence starting on line 38-40 to make it clear

Review standard P- value reporting for values less than 0.0001 (see line 41)

General comment: Tables and figures should have been included in the document for ease of corroboration of the figures, then if the document proceeds to productions then can be added and Figures, tables and supporting documents in the required file versions

L230, 275: review grammar. Sentence should not start with number symbol.

Authors to review publication policy on supporting information, supplemental material and figures and tables with data.

Some of the references are too old e.g ref 5, 6

Reviewer #2: Summary: The paper reports the results of a meta-analysis of 46 studies involving 210,960 participants on the relationship between vision impairment and measures of difficulties in daily life. The paper focuses on two broad measures: 1) basic activities of daily living (ADL), which refer to fundamental skills for daily self-care like eating and bathing, and 2) instrumental ADL, which refers to more complex activities like managing finances and preparing meals. The meta-analysis consisted of random effects models to calculate the pooled effect size for the correlation between vision impairment and the target measures. The results were a large positive correlation between vision impairment and difficulties in daily life, which suggests that worsening vision results in greater difficulties in daily life. Although the results of this study are correlational, the meta-analytic approach strengthens the claim that the relationship between vision impairment and difficulties with daily life is real and that causal factors and interventions should be investigated.

Opinion: This paper is a systematic review. The methods ensure a comprehensive and unbiased sampling of the existing literature, and the results have not been published elsewhere. The article adheres to appropriate reporting guidelines and standards for data availability. The article is presented in an intelligible fashion and the conclusions are clearly presented and are supported by the data. Most of the methods/statistics are performed to a high technical standard and described in sufficient detail. However, my main concern is about the age variable and specifically how the authors attempted to evaluate the role of age using the subgroups analysis (see review). Age is likely a mediator between visual impairment and difficulty in daily life, and the current approach taken by the authors does not adequately rule this out this mediating relationship. If this analysis is included and the current results stand, then this paper should be published.

Major:

1. I am particularly interested in the age variable. I expect that older individuals and those with disabilities have much higher scores on basic ADL, regardless of whether impaired vision is their disability. Nonetheless, vision often gets worse with age which means that age may be a mediating variable. The subgroup analysis (pp 279-307) on age attempts to address this issue, but the approach was quite odd. The authors compared the pooled effect in individuals aged 18-99 (presumably the entire sample) but also in individuals aged 65 year or more, and found no difference between the groups. Based on the reporting, it appears that the first group contains the members of the second group. At the very least, this needs to be better explained and justified. I would like to see a true subgroup analysis comparing different age groups. The best approach, however, would be for age to be included as a factor in the random effects model.

Minor:

1. Consider addition of summary stats in results. Basic ADL describes activities associated with basic physical needs like eating or toileting, whereas instrumental ADL involves more complicated cognitive tasks, such as balancing a checkbook or following a recipe. These two measures are very different in that individuals who have high difficulty with basic ADL are unable, or nearly unable, to physically care for themselves at all. In contrast, for IADL someone could have high difficulties with certain tasks like meal preparation but be entirely able to care for themselves. I think it is important for the authors to include some data about the range of these scores, and how these scores themselves relate to each other and to other variables such as age. This would give insight into the general abilities of the subjects without considering visual ability.

2. Clarify lines 283-288: Subgroup analysis based on the different assessments of vision impairment showed a higher non-significant difficulty in IADL. What is “non-significant difficulty”? In the following sentence you say “significant difficulty”. Non-significant versus significant difficulties need to be better defined, and the term significant should be avoided here where you are also using the same term to describe the results of significance tests.

3. Edit awkward/extraneous lines 330-333: The ROC results from another study should not be reported in the Discussion, and the way the ROC results are reported is vague and uninformative. I suggest complete removal of the ROC results and a higher-level description of the relationship between visual acuity threshold and IADL.

4. Explain lines 352-355: Statistical heterogeneity does not increase just because there is a large number of studies in the meta-analysis, but rather that there were variations in the original study designs such that there is more random error than just measurement error. This is obvious from the details in Table 1. The authors somewhat address this issue by performing a subgroup analysis on the different measures that make up the broad ADL and IADL category measures. However, it would be helpful for the authors to spend a bit more time discussing these differences rather than brushing it off as “expected given the large number of studies”.

6. PLOS authors have the option to publish the peer review history of their article (what does this mean?). If published, this will include your full peer review and any attached files.

Reviewer #1: No

Reviewer #2: No

---

## [Author Response · Author response to Decision Letter 0]

16 Dec 2024

Dear Editor,

Thank you for your feedback and for coordinating the review process for our manuscript. We appreciate the time and effort that both you and the reviewers have invested in improving the quality of our work. We are pleased to hear that our manuscript has been accepted pending minor revisions. 

We agree with Reviewer 2 and your suggestion that age is an important moderator of both vision impairment, ADL and IADL. To address this, we performed a meta-regression analysis to evaluate the effect of age on ADL and IADL among individuals with visual impairment. We have revised the Methods, Results, and Abstract sections to clearly describe the meta-regression analysis and explicitly discuss the potential mediating role of age. These results will be incorporated into the revised manuscript along with a discussion of their implications.

Reviewer’s #1 comments

Generally a well thought, well written manuscript.

Dear Reviewer,

Thank you very much for your thoughtful review of our manuscript. We hope you will find our response to your comments to be satisfactory as they helped us to significantly improve the manuscript. Please see below for a description and/or discussion of how we addressed each point raised.

Comment # Comments and Responses 

1

 Comment:

Abstract

Background: Doesn’t reflect scholarly inquest or bring out the problem that the researchers sought to study. 

Review the grammar of sentence starting on line 38-40 to make it clear.

Review standard P- value reporting for values less than 0.0001 (see line 41).

Response: 

We have revised the abstract to better reflect the research problem, clarified the grammar in the specified sentence, and ensured standard reporting of p-values less than 0.0001 as follows.

Background: Vision impairment is a common disability that poses significant challenges to individuals’ ability to perform activities essential for independent living, including activities of daily living (ADL) and instrumental activities of daily living (IADL). Despite extensive research, the extent and nature of these associations remain unclear, particularly across varying levels and types of vision impairment. 

Methods: We conducted a systematic review of relevant literature from the inception of the databases to February 2024, using electronic database searches, including PubMed, MEDLINE (Ovid), EMBASE, Cochrane CENTRAL, and CDSR.

2 Comment:

General comment: Tables and figures should have been included in the document for ease of corroboration of the figures, then if the document proceeds to productions then can be added and Figures, tables and supporting documents in the required file versions.

Response: 

Based on the reviewer’s insightful comment, all figures have now been included in the main manuscript for ease of corroboration. Additionally, funnel plots and other supplementary figures have been provided in the supplementary file as recommended. 

3 Comment:

L230, 275: review grammar. Sentence should not start with number symbol.

Response: 

We have revised the sentences at Lines 230 and 275 to ensure proper grammar and avoid starting with a number symbol.

4 Comment:

Authors to review publication policy on supporting information, supplemental material and figures and tables with data. Some of the references are too old e.g ref 5, 6.

Response: 

We have reviewed the publication policy regarding supporting information, supplemental material, and the inclusion of figures and tables with data to ensure compliance. Additionally, we have updated some of the older references, to incorporate more recent and relevant citations.

Reviewer’s #2 comments

The paper reports the results of a meta-analysis of 46 studies involving 210,960 participants on the relationship between vision impairment and measures of difficulties in daily life. The paper focuses on two broad measures: 1) basic activities of daily living (ADL), which refer to fundamental skills for daily self-care like eating and bathing, and 2) instrumental ADL, which refers to more complex activities like managing finances and preparing meals. The meta-analysis consisted of random effects models to calculate the pooled effect size for the correlation between vision impairment and the target measures. The results were a large positive correlation between vision impairment and difficulties in daily life, which suggests that worsening vision results in greater difficulties in daily life. Although the results of this study are correlational, the meta-analytic approach strengthens the claim that the relationship between vision impairment and difficulties with daily life is real and that causal factors and interventions should be investigated.

Opinion: This paper is a systematic review. The methods ensure a comprehensive and unbiased sampling of the existing literature, and the results have not been published elsewhere. The article adheres to appropriate reporting guidelines and standards for data availability. The article is presented in an intelligible fashion and the conclusions are clearly presented and are supported by the data. Most of the methods/statistics are performed to a high technical standard and described in sufficient detail. However, my main concern is about the age variable and specifically how the authors attempted to evaluate the role of age using the subgroups analysis (see review). Age is likely a mediator between visual impairment and difficulty in daily life, and the current approach taken by the authors does not adequately rule this out this mediating relationship. If this analysis is included and the current results stand, then this paper should be published.

Dear Reviewer,

Thank you for your thorough evaluation of our manuscript and for recognizing the rigor of our systematic review, adherence to reporting standards, and intelligibility of the presentation. We greatly appreciate your constructive feedback regarding the role of age as a potential mediator in the relationship between visual impairment and difficulties in daily life.

You raised an important point about the mediating role of age. To address this, we performed a meta-regression analysis to evaluate the effect of age on ADL and IADL among individuals with visual impairment. We have revised the Methods, Results, and Abstract sections to clearly describe the meta-regression analysis and explicitly discuss the potential mediating role of age.

Comment # Comments and Responses 

1

 Comment:

Major:

1. I am particularly interested in the age variable. I expect that older individuals and those with disabilities have much higher scores on basic ADL, regardless of whether impaired vision is their disability. Nonetheless, vision often gets worse with age which means that age may be a mediating variable. The subgroup analysis (pp 279-307) on age attempts to address this issue, but the approach was quite odd. The authors compared the pooled effect in individuals aged 18-99 (presumably the entire sample) but also in individuals aged 65 year or more, and found no difference between the groups. Based on the reporting, it appears that the first group contains the members of the second group. At the very least, this needs to be better explained and justified. I would like to see a true subgroup analysis comparing different age groups. The best approach, however, would be for age to be included as a factor in the random effects model.

Response: 

We acknowledge that our initial subgroup analysis (ages 18–99 vs. ≥65 years) may have created overlap between the groups, which limits its interpretability. To address this concern:

1. We performed a meta-regression analysis to include age as a continuous predictor in the random-effects model. This allowed us to directly assess the relationship between age and the outcomes (ADL/IADL) across all included studies. The results revealed a significant negative association between age and both ADL and IADL performance, as detailed in our revised manuscript.

2. We have revised the Methods and Results sections to clarify our approach.

We believe that the meta-regression approach better addresses your concern and provides a more robust understanding of age as a contributing factor. Thank you again for this constructive feedback, which has greatly improved the manuscript.

Based on the reviewer’s insightful comment, we revised the manuscript as follows:

Abstract:

Meta-regression analysis indicated that for every one-year increase in age, ADL performance decreased by an average of 0.0147 units (p < 0.001), while IADL performance declined at a slower rate of 0.0047 units/year (p = 0.031).

Methods:

To evaluate the potential impact of age on the relationship between vision impairment and ADL or IADL, a random-effects meta-regression analysis was conducted. The dependent variable was the Fisher z-transformed correlation coefficient, and age was used as the moderator (independent variable) in the analysis, employing the restricted maximum likelihood (REML) approach.

Results:

3.5. Meta-regression analysis 

To explore the potential sources of heterogeneity and examine the moderating role of age on the relationship between vision impairment and the performance of individuals in ADL and IADL, meta-regression analysis was performed. Age was used as the primary moderator variable in the regression model. The analysis revealed a significant negative association between age and both ADL and IADL performance. For ADL, the slope was -0.0147 (95% CI: -0.0179 to -0.0116, p < 0.001; S2a Fig in Supporting information), indicating that for every one-year increase in age, ADL performance decreased by an average of 0.0147 units. For IADL, the slope was -0.0047 (95% CI: -0.0088 to -0.0005, p = 0.031; S2b Fig in Supporting information), suggesting that IADL performance also declined with age, though the effect size was smaller compared to ADL. These findings highlight the significant moderating impact of age on functional performance, particularly in individuals with vision impairment.

2 Comment:

Minor:

Consider addition of summary stats in results. Basic ADL describes activities associated with basic physical needs like eating or toileting, whereas instrumental ADL involves more complicated cognitive tasks, such as balancing a checkbook or following a recipe. These two measures are very different in that individuals who have high difficulty with basic ADL are unable, or nearly unable, to physically care for themselves at all. In contrast, for IADL someone could have high difficulties with certain tasks like meal preparation but be entirely able to care for themselves. I think it is important for the authors to include some data about the range of these scores, and how these scores themselves relate to each other and to other variables such as age. This would give insight into the general abilities of the subjects without considering visual ability.

Response: 

Thank you for your insightful suggestion. Unfortunately, due to the nature of the data available, we were unable to perform the specific analyses you recommended. However, we conducted a meta-regression analysis incorporating age as a continuous predictor in the random-effects model. This allowed us to assess the relationship between age and the outcomes (ADL/IADL) across all included studies. The results revealed a significant negative association between age and both ADL and IADL performance, and these findings have been included in the revised manuscript. We hope this analysis addresses your concern.

3 Comment:

Clarify lines 283-288: Subgroup analysis based on the different assessments of vision impairment showed a higher non-significant difficulty in IADL. What is “non-significant difficulty”? In the following sentence you say “significant difficulty”. Non-significant versus significant difficulties need to be better defined, and the term significant should be avoided here where you are also using the same term to describe the results of significance tests.

Response: 

We have revised the sentence to clarify the distinction between "non-significant" and "significant" findings. The updated sentence now reads:

Subgroup analysis based on the different assessments of vision impairment showed higher odds of IADL in studies that used self-reported assessment (OR = 2.19, 95% CI 1.85 - 2.61, p = 0.00001, I2 = 89%, p = 0.00001) compared with studies that objectively assessed vision impairment (OR = 1.83, 95% CI 1.46 - 2.28, p = 0.00001, I2 = 97%, p = 0.00001). However, the test for subgroup difference was not statistically significant (Chi2 = 1.65, p = 0.20).

This revision clarifies that although the difficulty in IADL was higher in the self-reported assessment subgroup, the difference between the subgroups was not statistically significant.

4 

5 Comment:

Edit awkward/extraneous lines 330-333: The ROC results from another study should not be reported in the Discussion, and the way the ROC results are reported is vague and uninformative. I suggest complete removal of the ROC results and a higher-level description of the relationship between visual acuity threshold and IADL.

Response: 

We have removed the section discussing the ROC results, as recommended.

 Comment:

Explain lines 352-355: Statistical heterogeneity does not increase just because there is a large number of studies in the meta-analysis, but rather that there were variations in the original study designs such that there is more random error than just measurement error. This is obvious from the details in Table 1. The authors somewhat address this issue by performing a subgroup analysis on the different measures that make up the broad ADL and IADL category measures. However, it would be helpful for the authors to spend a bit more time discussing these differences rather than brushing it off as “expected given the large number of studies”.

Response: 

We have revised the text to provide a clearer explanation of the statistical heterogeneity observed in our meta-analysis. The revised version now reads:

First, significant statistical heterogeneity was observed in the results, which can be attributed to variations in patient characteristics, vision assessment methods and definition, and the different ADL and IADL assessment tools used across studies. This heterogeneity arises not only from measurement error but also from the inherent differences in study designs, as detailed in Table 1. While we addressed this issue through subgroup analysis based on the different ADL and IADL measures, we have further discussed these differences to clarify the sources of heterogeneity and their potential impact on the findings.

---

## [Editor Report · Decision Letter 1]

30 Dec 2024

Vision impairment and associated daily activity limitation: A systematic review and meta-analysis

PONE-D-24-38439R1

Dear Dr. Rahmati,

We’re pleased to inform you that your manuscript has been judged scientifically suitable for publication and will be formally accepted for publication once it meets all outstanding technical requirements.

Kind regards,

Katya Numbers, PhD, M.S., B.S.

Academic Editor

PLOS ONE

Additional Editor Comments (optional):

Reviewer 1 originally had very minor comments/suggestions and originally suggested Accept, where Reviewer 2 suggested Major Revisions, mostly centered on whether there should be an additional stratified analysis based on age. The authors have conducted a new meta-regression with age as a predictor. Age was significantly negatively associated with IADL and ADL scores, as expected, and the authors and I believe this additional analysis and information strengthens the paper. I am happy to suggest the article should be Accepted in its revised form, particularly as all other queries have also been addressed and were relatively minor. - Katya
---

## [Editor Report · Acceptance letter]

24 Jan 2025

PONE-D-24-38439R1 

PLOS ONE

Dear Dr. Rahmati, 

I'm pleased to inform you that your manuscript has been deemed suitable for publication in PLOS ONE. Congratulations! Your manuscript is now being handed over to our production team.

Kind regards, 

on behalf of

Dr. Katya Numbers 

Academic Editor

PLOS ONE